# Sparse Learning for State Space Models on Mobile

**Xuan Shen**[1][*] **Hangyu Zheng**[2][*] **Yifan Gong**[1] **Zhenglun Kong**[3] **Changdi Yang**[1]
**Zheng Zhan**[1] **Yushu Wu**[1] **Xue Lin**[1] **Yanzhi Wang**[1] **Pu Zhao**[1] **Wei Niu**[2][†]
[1]Northeastern University   [2]University of Georgia   [3]Harvard University
{shen.xu}@northeastern.edu  {hz85760, wniu}@uga.edu

## Abstract

Transformer models have been widely investigated in different domains by providing long-range dependency handling and global contextual awareness, driving the development of popular AI applications such as ChatGPT, Gemini, and Alexa. State Space Models (SSMs) have emerged as strong contenders in the field of sequential modeling, challenging the dominance of Transformers. SSMs incorporate a selective mechanism that allows for dynamic parameter adjustment based on input data, enhancing their performance. However, this mechanism also comes with increasing computational complexity and bandwidth demands, posing challenges for deployment on resource-constraint mobile devices. To address these challenges without sacrificing the accuracy of the selective mechanism, we propose a sparse learning framework that integrates architecture-aware compiler optimizations. We introduce an end-to-end solution–$\mathbf{C}_4^n$ kernel sparsity, which prunes $n$ elements from every four contiguous weights, and develop a compiler-based acceleration solution to ensure execution efficiency for this sparsity on mobile devices. Based on the kernel sparsity, our framework generates optimized sparse models targeting specific sparsity or latency requirements for various model sizes. We further leverage pruned weights to compensate for the remaining weights, enhancing downstream task performance. For practical hardware acceleration, we propose $\mathbf{C}_4^n$-specific optimizations combined with a layout transformation elimination strategy. This approach mitigates inefficiencies arising from fine-grained pruning in linear layers and improves performance across other operations. Experimental results demonstrate that our method achieves superior task performance compared to other semi-structured pruning methods and achieves up-to $7\times$ speedup compared to llama.cpp framework on mobile devices.

## 1 Introduction

Recent research advancements have significantly heightened interest in State Space Models (SSMs). Building on the foundation of the Kalman filter model (kal, 1960), SSMs have been further improved to address long-range dependencies with parallel training. Works (Gu et al., 2021a;b; 2022; Gupta et al., 2022) propose SSM-based models designed to process sequence data across a variety of tasks and modalities. Recent work, Mamba (Gu & Dao, 2023b), integrates time-varying parameters into the SSM, enabling the model to selectively propagate or forget information. Additionally, Mamba introduces a hardware-aware parallel algorithm designed to accelerate training and inference. Compared to quadratic attention, which becomes prohibitively expensive with longer sequence lengths, Mamba's subquadratic-time architecture is more efficient and better suited for handling long sequences. Mamba's exceptional scaling performance highlights its potential as an effective alternative to the Transformer model (Vaswani et al., 2017) for generative language modeling tasks.

To make advanced language processing accessible to more people and address privacy concerns, deploying Mamba models on mobile devices is a promising strategy to improve the accessibility and usability of SSMs. A report indicates that by the end of 2020, nearly 6 billion smartphones were in

---

[*]Equal Contribution

[†]Corresponding Author

use worldwide, experiencing an annual growth rate of 4% (Samsung), and demonstrating increasing processing capabilities (Huynh et al., 2017; Xu et al., 2018; Chen et al., 2023; Zhan et al., 2021; Wu et al., 2022; Li et al., 2022; 2023a; Shen et al., 2024c; Niu et al., 2024a). Deploying Mamba models on mobile devices ensures functionality in offline scenarios and reduces reliance on costly cloud services. However, the hardware-aware parallel algorithm in Mamba models is specifically optimized for GPUs, similar optimizations for mobile devices have yet to be explored. The computational complexity in Mamba poses significant challenge for resource-constraint mobile devices, where bandwidth is limited compared to desktop-level GPUs (Huynh et al., 2017; Lee et al., 2019). Moreover, the projection mechanism within Mamba demands high throughput memory. Unlike powerful server platforms, mobile devices typically have less memory size and are constrained by limited battery capacity, which restricts their computing performance and memory bandwidth. For instance, state-of-the-art framework – llama.cpp (contributors, 2023a) takes over **2.5s** to generate one token with Mamba-2.8B model on a high-end mobile device (Qualcomm, 2017), highlighting the need for further optimization to make Mamba models viable for mobile deployment.

To tackle the mentioned challenges on mobiles, we propose a sparse learning framework that incorporates architecture-aware compiler optimizations for the acceleration of Mamba models on mobile devices. Inspired by the modern hardware architecture of Single Instruction Multiple Data (SIMD) (Cypher & Sanz, 1989; Mitra et al., 2013; Khorasani et al., 2015) units, which are optimized to load and process four-element vectors in parallel. We focus on investigating the sparse patterns within every four elements to improve the hardware efficiency with the support of our compiler optimization. We first introduce the $\mathbf{C}_4^n$ kernels which prune $n$ elements from every four contiguous weights. We further propose a sparse learning framework to thoroughly optimize the pruning strategy for these kernels, i.e., determine the value of $n$ for each kernel (four-element vector) and the corresponding pruned $n$ elements. To preserve task performance while applying sparsity and achieving significant acceleration, we profile the effectiveness loss (as an accuracy predictor), sparsity loss and latency loss for each choice of $n$ for each kernel. With the detailed profiling, given the specific sparsity or latency goals, our sparse learning framework targets to learn a mask to choose a value of $n$ for each kernel, so that the accuracy loss is minimized while satisfying the sparsity/latency constraint. In detail, to render the sparse learning process differentiable, we define the pruning strategy (mask) through probabilities assigned to different kernels for every set of four consecutive weights throughout the entire model. Finally, we introduce a compensation algorithm to rectify the remaining weights by utilizing the pruned weights, optimizing overall model functionality. The compensation is mainly based on the classic OBS update (Hassibi et al., 1993; Singh & Alistarh, 2020; Frantar et al., 2021; Zhao et al., 2024) for the weight reconstruction, leveraging calibration with only 128 training samples. Regarding hardware acceleration, we propose a unique design for $\mathbf{C}_4^n$-specific optimizations that includes weight reordering and an efficient method for storing sparse weights. For other operators with intensive memory movement, we design a layout transformation elimination to decrease the bandwidth demands without the need for data layout changes. These optimizations are crucial to mitigate the performance degradation associated with fine-grained $\mathbf{C}_4^n$ pruning versus structured pruning or dense configurations. Experiments show that our algorithmic approaches can achieve better performance with the same sparsity on different scales of Mamba models compared to other semi-structure pruning methods with fixed sparsity patterns. Specifically, we reduce the perplexity from 212.9 to 28.97 and enhance the accuracy from 35.6% to 41.0% on Mamba-130M model compared to Wanda (Sun et al., 2023) with 2:4 pattern. Our comprehensive ablation study demonstrates the effectiveness of our mixed kernel design and the compensation methods. We implement the sparse model with our proposed kernels on mobile devices and achieve a practical on-device speedup of up to $7\times$ compared to llama.cpp. We summarize our contribution:

**1.** We design a special kernel $\mathbf{C}_4^n$ and with a set of comprehensive compiler optimizations, including $\mathbf{C}_4^n$-specific optimizations and layout transformation elimination strategy on mobile devices.

**2.** We propose the sparsity-oriented and/or latency-oriented sparse learning framework to explore the optimal pruning strategy with the proposed kernels for Mamba models.

**3.** We propose the weight compensation algorithm for the rectification of the sparse model weights by calibrating with only 128 samples, thereby further enhancing the model effectiveness.

**4.** Experiments show that our framework can achieve better task performance than other semi-structure pruning methods and achieve pratical on-device speedup up to $7\times$ compared to llama.cpp.

## 2 RELATED WORK

### 2.1 STATE SPACE MODELS

The work (Gu et al., 2021a) initially models long sequences using structured state spaces rather than Transformers (Vaswani et al., 2017) or Convolutional Neural Networks (CNNs), sparking interest in exploring state space models. The memory usage in Transformer increases with the context length, making it difficult to efficiently process long-context windows or multiple parallel batches without substantial hardware resources. Recently, the work (Fu et al., 2022) fills the performance gap between SSMs and Transformers in language modeling, and Mamba (Gu & Dao, 2023a) introduces a new Mamba structure as the general sequence model backbones. Mamba introduces an input-dependent selection mechanism into SSMs and benefits from linear scaling in sequence length, surpassing traditional Transformers across multiple model sizes. However, previous study Zhan et al. (2024a) does not fully investigate the redundancy inherent in SSMs and hardware acceleration on resource-constraint mobile devices, leaving this research area largely under-explored.

### 2.2 DNN INFERENCE ACCELERATION ON MOBILE

As machine learning applications (Li et al., 2023b; 2024a; Zhang et al., 2022; Yang et al., 2023) on mobile devices continue to grow, there is a strong emphasis on optimizing frameworks of deep neural network (DNN) inference on mobile. Efforts such as works including (Han et al., 2016; Niu et al., 2020) have primarily focused on accelerating traditional CNNs. General inference frameworks that support both server and mobile platforms for different neural networks, such as (Niu et al., 2021; 2024b) offer advanced features including operator fusion, memory planning, shape inference, quantization, and tensor offloading. More recently, there has been a trend towards working with large models like llama.cpp (contributors, 2023a), exLLaMa (exllama contributors, 2023), MLC-LLM (contributors, 2023b), and fastLLM (The fastllm contributors, 2023). Yet many of these efforts either overlook model efficient techniques (Shen et al., 2024d;e;a;b; 2023; 2022; Li et al., 2024b; Zhan et al., 2024b) or fail to support SSMs on mobile platforms.

## 3 MOTIVATION AND BACKGROUND

To showcase the superior efficiency and mobile-friendliness of Mamba models over Transformers, we conduct a throughput comparison between them. As shown in Figure 1, by comparing the model size and throughput (tested on a Oneplus 11 mobile phone) under the same configuration (such as the same batch size and input sequence length), Mamba models can achieve a higher throughput with a model size similar or even larger than the Transformer models from various LLM families (Mehta et al., 2024; Yang et al., 2024; Le Scao et al., 2023), leading to a better trade-off between throughput and model size.

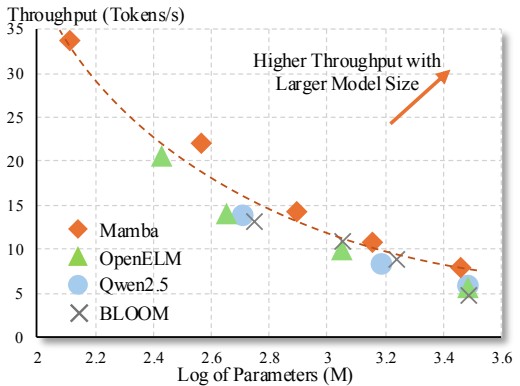

Figure 1: Throughput v.s. Log of Parameters (M)

Besides, in practice, under the same memory usage, Mamba models typically can use a larger batch size, resulting in 4-5× higher inference throughput than a Transformer of similar size. The reason is that, unlike Transformers, Mamba models don't require the KV cache (Gu & Dao, 2023a), reducing memory usage and allowing for larger batch sizes with higher throughput.

Furthermore, because of Mamba's ability to selectively remember the relevant token while ignoring everything else in between, it can even use extremely long context with length up to 1M. On induction heads task (Olsson et al., 2022), it generalizes perfectly to million-length sequences, or 4000× longer than it saw during training, while no other method goes beyond 2× (Gu & Dao, 2023a).

Due to the efficiency and mobile-friendliness of Mamba models over Transformers, we focus on optimizing Mamba models for superior on-mobile performance.

## 4 PRELIMINARY

State Space Models (SSMs) are sequential models that can map a 1-dimensional function or sequence $x(t) \in \mathbb{R}$ to the output sequence $y(t) \in \mathbb{R}$ through a hidden state $h(t) \in \mathbb{R}^N$ as follows,

$$
\begin{aligned}
h'(t) &= \mathbf{A}h(t) + \mathbf{B}x(t), \\
y(t) &= \mathbf{C}h(t),
\end{aligned}
\tag{1}
$$

where $N$ denotes the representation number, $\mathbf{A} \in \mathbb{R}^{N \times N}$ is evolution parameter, $\mathbf{B} \in \mathbb{R}^{N \times 1}$ and $\mathbf{C} \in \mathbb{R}^{1 \times N}$ are projection parameters.

The Mamba model (Gu & Dao, 2023b) represents the discrete version of the continuous system for SSMs and incorporates a timescale parameter $\Delta$ to facilitate the transformation of continuous parameters with the zero-order hold (ZOH) as follows,

$$
\begin{aligned}
\overline{\mathbf{A}} &= \exp(\Delta\mathbf{A}), \\
\overline{\mathbf{B}} &= (\Delta\mathbf{A})^{-1}(\exp(\Delta\mathbf{A}) - \mathbf{I}) \cdot \Delta\mathbf{B}.
\end{aligned}
\tag{2}
$$

After getting the discretized $\overline{\mathbf{A}}$ and $\overline{\mathbf{B}}$, the discretization of Equation (1) can be rewritten as follows,

$$
\begin{aligned}
h_t &= \overline{\mathbf{A}}h_{t-1} + \overline{\mathbf{B}}x_t, \\
y_t &= \mathbf{C}h_t.
\end{aligned}
\tag{3}
$$

At last, the Mamba model computes the output through a global convolution as follows,

$$
\begin{aligned}
\overline{\mathbf{K}} &= (\mathbf{C}\overline{\mathbf{B}}, \mathbf{C}\overline{\mathbf{A}}\overline{\mathbf{B}}, \cdots, \mathbf{C}\overline{\mathbf{A}}^{\mathbf{L}-1}\overline{\mathbf{B}}), \\
\mathbf{y} &= \mathbf{x} * \overline{\mathbf{K}}
\end{aligned}
\tag{4}
$$

where $y$ denotes the output sequence, $L$ denotes the length of the input sequence $\mathbf{x}$ and $\overline{\mathbf{K}} \in \mathbb{R}^L$ denotes one structured convolutional kernel.

## 5 METHODOLOGY

In this section, we start by exploring the design philosophy for sparse patterns on mobile devices. Next, we present a sparse learning framework aimed at optimizing the model's sparse structure through effective loss, sparsity loss, and latency loss for each layer. We then introduce a compensation method that leverages pruned weights to further optimize the remaining weights. Finally, we illustrate a set of comprehensive compiler-enabled optimizations for proposed kernels.

### 5.1 SPARSE KERNEL DESIGN

**Rationality of our sparse kernels** We split the weights into multiple non-overlapping groups and each group has 4 adjacent weights. Our kernel is designed as $\mathbf{C}_4^n$, which removes $n$ elements from every group with four adjacent weights. This approach is inspired by the architecture of modern hardware's Single Instruction Multiple Data (SIMD) units that process groups of four elements at once (Cypher & Sanz, 1989; Mitra et al., 2013; Khorasani et al., 2015). Utilizing SIMD's ability to handle vectors of four elements in parallel boosts computational efficiency and performance (Chen et al., 2018; Alibaba, 2020). By tailoring our pruning kernels to fit the SIMD architecture, we enhance the inference computations to fully leverage hardware capabilities, leading to higher efficiency with faster speed. Details are further explained in Section 5.6.

**Latency profiling** In our sparse kernels with $\mathbf{C}_4^n$, $n$ can be different values leading to various sparsity and latency. To get a deeper understanding for our $\mathbf{C}_4^n$ and $n$, we collect latency data from various kernels for our next sparse

Table 1: Latency profiling for different kernels.

| Kernel | $\mathbf{C}_4^0$ | $\mathbf{C}_4^1$ | $\mathbf{C}_4^2$ | $\mathbf{C}_4^3$ | $\mathbf{C}_4^4$ |
|---|---|---|---|---|---|
| Sparsity | 0% | 25% | 50% | 75% | 100% |
| Latency (ms) | 37.14 | 29.66 | 22.71 | 19.05 | 5.74 |

learning process. A synthesized model with random weights is utilized for profiling on a mobile CPU since the weight values barely affect latency. The representative profiling results are shown in Table 1. We can establish the latency function $\mathcal{F}_T(\cdot)$ using Table 1. This function is then used in the latency loss, as detailed in Section 5.3.

## 5.2 EFFECTIVENESS LOSS

Next we perform sparse learning for the SSM model to apply our sparse kernels for the whole model. To mitigate the accuracy degradation, it is essential to develop an accuracy predictor for our sparse learning, given that evaluating model performance on the full testing dataset is resource-intensive and not conducive to model generalization. To address this problem, we introduce an effectiveness loss $\mathcal{L}_E$ based on the weight removal error, as the accuracy predictor. Considering the computational constraints, a global analysis of weight removal error presents significant challenges. Consequently, a layer-wise investigation emerges as a viable approach under these limitations.

Given the input $\mathbf{X} \in \mathbb{R}^{D_{in} \times L \times B}$ for one layer with weight $\mathbf{W} \in \mathbb{R}^{D_{out} \times D_{in}}$, where $B$ denotes the batch size, $L$ is the sequence length, and $D_{in}/D_{out}$ denote the input/output dimensions, to mitigate the performance loss, the difference of outputs before and after pruning is minimized as follows,

$$\min_{\mathbf{M}, \widehat{\mathbf{W}}} \quad L = \|\mathbf{WX} - (\widehat{\mathbf{W}} \odot \mathbf{M})\mathbf{X}\|_2^2, \tag{5}$$

where $\|\cdot\|_2^2$ denotes the $\ell_2$ norm, $\mathbf{M} \in \mathbb{R}^{D_{out} \times D_{in}}$ is the sparse mask indicating the pruned locations, $\odot$ denotes the element-wise multiplication, and $\widehat{\mathbf{W}}$ denotes the optimized weights. To make the problem tractable, we assume that the sparse mask is given and fixed during the optimization, and we can have the following solution with detailed proof in Appendix A.

**Theorem 5.1.** *The optimal solution to Problem (5) with a fixed $\mathbf{M}$ can be obtained by the following,*

$$\widehat{\mathbf{W}}^* = \mathbf{W} - \left( \sum_i \mathbf{e}_{q_i} \mathbf{e}_{q_i}^T \mathbf{W} \mathbf{M}_i [\mathbf{M}_i^T (2\mathbf{XX}^T)^{-1} \mathbf{M}_i]^{-1} \mathbf{M}_i^T \right) \times (2\mathbf{XX}^T)^{-1}. \tag{6}$$

*and the minimal loss can be expressed as*

$$L^* = \frac{1}{2} \sum_i \mathbf{e}_{q_i}^T \mathbf{W} \mathbf{M}_i [\mathbf{M}_i^T (2\mathbf{XX}^T)^{-1} \mathbf{M}_i]^{-1} \mathbf{M}_i^T \mathbf{W}^T \mathbf{e}_{q_i}. \tag{7}$$

*The sparse locations are distributed in a total of $k$ rows in $\mathbf{W}$, and their row indices can be denoted by $\mathbf{e}_{q_i} \in \{0, 1\}^{D_{out} \times 1}, i = 1, ..., k$, where $\mathbf{e}_{q_i}$ is a one-hot vector with the $q_i^{th}$ element as 1 and all others as 0. There are $k_i$ elements pruned in the $q_i^{th}$ row and their indices can be represented by $\mathbf{e}_{p_j - q_i} \in \{0, 1\}^{D_{in} \times 1}, j = 1, ..., k_i$, which are also one-hot vectors similar to $\mathbf{e}_{q_i}$. $\mathbf{M}_i \in \mathbb{R}^{D_{in} \times k_i}$, where the $j^{th}$ column of $\mathbf{M}_i$ is $[\mathbf{M}_i]_{:,j} = \mathbf{e}_{p_j - q_i}, \forall j$.*

*Remark 5.2.* If $2\mathbf{XX}^T$ is not full rank with difficulties for the inversion $(2\mathbf{XX}^T)^{-1}$, the dampening technique is adopted to compute $(2\mathbf{XX}^T + \gamma\mathbf{I})^{-1}$ instead, with $\gamma$ as the dampening ratio.

*Remark 5.3.* Although $\mathbf{M}$ is fixed during the optimization, the optimal $\mathbf{M}$ can be obtained by minimizing the loss in Equation (7), i.e., each mask corresponds to a loss in Equation (7) and the mask with the minimal loss is the optimal one. But it typically incurs unaffordable complexity to find the optimal mask by comparing the losses of all masks.

**Kernel-wise effectiveness loss** Based on the optimal loss, we can obtain the effectiveness loss to estimate the weight removal error for each kernel. For example, for $\mathbf{C}_4^2$ which has 2 zero elements among 4 elements, there are 6 combinations to select 2 elements from 4, and we compute the loss for each combination following Equation (7). Then we sort the 6 losses and find out the minimal loss as the effectiveness loss with the corresponding two pruned elements. We can perform the same process for other cases such as $\mathbf{C}_4^3$ with 3 zeros among 4 weights. In this way, the effectiveness loss for all cases ($\mathbf{C}_4^0, \mathbf{C}_4^1, \mathbf{C}_4^2, \mathbf{C}_4^3, \mathbf{C}_4^4$) of each kernel can be obtained. Note that once the effectiveness loss is determined, the corresponding pruning locations or weights for each kernel are also obtained as other combinations lead to larger weight removal error.

**Effectiveness loss of the layer** For the weight subset $\mathbf{w}_{i,j} \in \mathbb{R}^{1 \times 4}$ with $i = 1, ..., D_{out}$ and $j = 1, ..., \lceil D_{in}/4 \rceil$ where $\lceil \cdot \rceil$ denotes rounding up to the nearest integer, the weight effectiveness $\mathbf{E}^{\mathbf{W}} \in \mathbb{R}^{D_{out} \times \lceil D_{in}/4 \rceil}$ according to the removal strategy $\mathbf{N}^{\mathbf{W}} \in \{0, 1, 2, 3, 4\}^{D_{out} \times \lceil D_{in}/4 \rceil}$ can be defined as follows,

$$[\mathbf{E}^{\mathbf{W}}]_{i,j} = \mathcal{F}_L(\mathbf{w}_{i,j}, [\mathbf{N}^{\mathbf{W}}]_{i,j}), \tag{8}$$

where $\mathcal{F}_L(\cdot, k)$ denotes the function to generate the effectiveness loss for each kernel with $k$ pruned elements according to Equation (7) and $[\mathbf{N}^{\mathbf{W}}]_{i,j}$ denotes the number of pruned elements in $\mathbf{w}_{i,j}$. Thus, we deliver the effectiveness loss $\mathcal{L}_E$ by accumulating weight removal errors of all kernels:

$$\mathcal{L}_E(\mathbf{E}^{\mathbf{W}}) = \sum_{i,j} [\mathbf{E}^{\mathbf{W}}]_{i,j}. \tag{9}$$

## 5.3 SPARSITY AND LATENCY LOSS

In our proposed sparse learning framework, the optimization of the sparse structure can be driven by accuracy or latency considerations. Thus, apart from the effectiveness loss, we include the additional sparsity loss $\mathcal{L}_S$ or the latency loss $\mathcal{L}_T$, to learn the sparse mask.

We first deliver the discrete sparsity $\mathbf{S^W} \in \mathbb{R}^{D_{out} \times \lceil D_{in}/4 \rceil}$ of the subset weight $\mathbf{w}_{i,j}$ with the removal strategy $\mathbf{N^W}$ as follows,

$$[\mathbf{S^W}]_{i,j} = [\mathbf{N^W}]_{i,j} / 4 \tag{10}$$

Subsequently, we define the sparsity loss $\mathcal{L}_S$ with the target sparsity ratio $S_t$ as follows,

$$\mathcal{L}_S(\mathbf{S^W}, S_t) = \begin{cases} S_t - \text{avg}(\mathbf{S^W}), & \text{if} \quad S_t > \text{avg}(\mathbf{S^W}) \\ 0, & \text{otherwise} \end{cases} \tag{11}$$

where $\text{avg}(\cdot)$ is the average function. We denote the discrete latency $\mathbf{T^W} \in \mathbb{R}^{D_{out} \times \lceil D_{in}/4 \rceil}$ of the sparse model using $\mathcal{F}_T(\cdot)$ defined in Section 5.1 with the look up table as follows,

$$[\mathbf{T^W}]_{i,j} = \mathcal{F}_T([\mathbf{N^W}]_{i,j}) \tag{12}$$

Then, the latency loss $\mathcal{L}_T$ can be defined with the target latency $T_t$ as follows,

$$\mathcal{L}_T(\mathbf{T^W}, T_t) = \begin{cases} \sum_{i,j} \mathbf{T^W} - T_t, & \text{if} \quad \sum_{i,j} \mathbf{T^W} > T_t \\ 0, & \text{otherwise} \end{cases} \tag{13}$$

## 5.4 SPARSE LEARNING

To optimize the removal strategy $[\mathbf{N^W}]_{i,j}$ for each sparse kernel $\mathbf{w}_{i,j}$ in one layer, we define the trainable probability mask as $\mathbf{M^W} \in \mathbb{R}^{D_{out} \times \lceil D_{in}/4 \rceil \times 5}$ with its subset $\mathbf{m}_{i,j}^{\mathbf{W}} \in \mathbb{R}^{1 \times 5}$ for each kernel. Each kernel has its unique effectiveness loss, sparsity loss and latency loss, corresponding to different pruned number, i.e, 0,1,2,3, or 4 with a total of 5 cases. The probability mask $\mathbf{m}_{i,j}^{\mathbf{W}}$ is designed to learn the number of pruned weights in $\mathbf{w}_{i,j}$ (choosing one from the 5 cases), and it is randomly initialized with the Gaussian distribution for the probability of removing 0,1,2,3, or 4 weights. We minimize the sum of the weight effectiveness and sparsity (or latency) losses with the sparse kernel selection denoted by the trainable mask $\mathbf{M^W}$ with softmax operation to learn the appropriate sparsity configuration for each kernel. The algorithm for sparse learning is presented in Algorithm 1, featuring $r_e$ as the effectiveness loss ratio and $r_t$ as the target loss ratio. The 'max_index' retrieves the index of the maximum value along a specified dimension. The output $\mathbf{N}_{output}$ generated by the algorithm represents the optimal sparse structure using the proposed kernel. Note that although we only find the number of pruned elements in each kernel for the sparse learning, the pruned locations corresponding to each choice of pruned number is already determined as discussed in Section 5.2. Thus, once $\mathbf{M^W}$ is learned, the pruned weights in each kernel are also obtained immediately.

---

**Algorithm 1:** Sparse Learning

---

**Input:** $\mathbf{W} \in \mathbb{R}^{D_{in} \times D_{out}}$, $r_e$, $r_t$, $S_t$ or $T_t$,
Create trainable mask $\mathbf{M^W} \in \mathbb{R}^{D_{out} \times \lceil D_{in}/4 \rceil \times 5}$.
Initialize all $\mathbf{m}_{i,j}^{\mathbf{W}} \subseteq \mathbf{M^W}$ with Gaussian distribution.
$[\mathbf{N}_u] = \{u\}^{D_{out} \times \lceil D_{in}/4 \rceil}$, $u = 0, 1, 2, 3, 4$
$[\mathbf{E}_u]_{i,j} = \mathcal{F}_L(\mathbf{w}_{i,j}, [\mathbf{N}_u]_{i,j})$, $u = 0, 1, 2, 3, 4, \forall i, j$
$\mathbf{S}_u = \mathbf{N}_u/4$, $u = 0, 1, 2, 3, 4$   or   $\mathbf{T}_u = \mathcal{F}_T([\mathbf{N}_u]_{i,j})$, $u = 0, 1, 2, 3, 4, \forall i, j$
**for** $e$ *in* $[1, steps]$ **do**
    $\mathbf{M}' = \text{softmax}(\mathbf{M^W}, \text{dim=-1})$
    $\mathcal{L} = r_e * \mathcal{L}_E(\text{sum}(\mathbf{M}' \odot [\mathbf{E}_0, \mathbf{E}_1, \mathbf{E}_2, \mathbf{E}_3, \mathbf{E}_4], \text{dim=-1}))$
       $+ r_t * \mathcal{L}_S(\text{sum}(\mathbf{M}' \odot [\mathbf{S}_0, \mathbf{S}_1, \mathbf{S}_2, \mathbf{S}_3, \mathbf{S}_4], \text{dim=-1}), S_t)$
      or
       $+ r_t * \mathcal{L}_T(\text{sum}(\mathbf{M}' \odot [\mathbf{T}_0, \mathbf{T}_1, \mathbf{T}_2, \mathbf{T}_3, \mathbf{T}_4], \text{dim=-1}), T_t)$
    Backward $\mathcal{L}$
    Update $\mathbf{M^W}$ with SGD
    Decay learning rate
**end**
**Output:** $\mathbf{N}_{output} = \text{max\_index}(\text{softmax}(\mathbf{M^W}), \text{dim=-1})$

---

## 5.5 COMPENSATION

After learning the sparse mask for each kernel in previous steps, we adjust the unpruned weights to offset the accuracy loss from pruning. Specifically, we update the unpruned weights according to Equation (6). Since $\mathbf{e}_{q_i}$ is a one-hot vector, $\mathbf{e}_{q_i} \times A$ only has non-zero values in the $q_i^{th}$ row with all zeros for all other rows. Thus, in Equation (6), each term with the index $i$ in the sum just computes the $q_i^{th}$ row in the outputs and the computation of the $q_i^{th}$ row does not affect the $q_s^{th}$ row, $\forall s \neq i$. Specifically, based on Theorem 5.1, we have the following,

$$[\widehat{\mathbf{W}}^*]_{q_i,:} = [\mathbf{W}]_{q_i,:} - \mathbf{e}_{q_i}^T \mathbf{W} \mathbf{M}_i [\mathbf{M}_i^T (2\mathbf{X}\mathbf{X}^T)^{-1} \mathbf{M}_i]^{-1} \mathbf{M}_i^T (2\mathbf{X}\mathbf{X}^T)^{-1} \tag{14}$$

Thus, we can perform optimal adjustments row by row. For each row, Equation (14) incorporates the pruned locations across all kernels in the same row and considers their connections, as shown by $[\mathbf{M}_i^T (2\mathbf{X}\mathbf{X}^T)^{-1} \mathbf{M}_i]^{-1}$.

## 5.6 HARDWARE IMPLEMENTATION

For Mamba models utilizing $\mathbf{C}_4^n$ pruning, our compiler aims for maximum hardware efficiency during inference by implementing advanced optimizations and code generation. The optimization strategy consists of two primary approaches: (1) *General optimizations*, which includes operator fusion, static memory planning, and parameter tuning for Mamba models. (2) $\mathbf{C}_4^n$-*specific optimizations* aimed at enhancing performance uniquely for the $\mathbf{C}_4^n$. Our compiler takes a model (in the form of computational graph) as input and generates source code output. Specifically, it produces C++/Assembly code for mobile CPUs and OpenCL code for mobile GPUs. Due to the space limitations, our general compiler optimizations details in Appendix B.1.

**$\mathbf{C}_4^n$-specific Optimizations** The irregular data access and computation patterns in pruned models significantly contribute to the inefficient execution of sparse DNNs (Sun et al., 2023; Frantar & Alistarh, 2023). Our compiler tackles this inefficiency through two strategies: (1) reordering the weights offline to remove computational uncertainty and balance load disparities in sparse computation. (2) developing an efficient sparse weight storage method that utilizes the kernel information to further reduce extra storage needs.

Figure 2 (a) and (b) illustrate examples of matrix multiplication with and without reordering. In Figure 2 (a), without reordering, the process encounters typical challenges of sparse matrix multiplication: control-flow divergence among threads, load imbalance across threads, and irregular memory access patterns. In Figure 2 (b), we initially group rows with similar numbers of non-zero elements in the weights to minimize load imbalance since different threads in a warp access contiguous rows. This approach reduces the number of idle threads caused by load imbalances. Within each row, we further reorder the same pattern of non-zero elements together to minimize the control-flow divergence. In Figure 2 (a) and (b), prior to reordering, the nested loop for computation induces if-else divergence, which is not hardware-friendly for parallel computing units, particularly in mobile GPUs. However, after reordering, we employ a regular for loop that can perform identical computations for identical patterns contiguously, thereby eliminating branch divergence.

After reordering the matrix, we store the model in a compact format using our $\mathbf{C}_4^n$-specific format, named $\mathbf{C}_4^n$ Compact as shown in Figure 2 (c). Unlike CSR (Buluç et al., 2009), which only eliminates zero weights, $\mathbf{C}_4^n$ Compact achieves a higher compression ratio through a hierarchical index structure that removes redundant column indices resulting from pruning with $\mathbf{C}_4^n$. This method efficiently conserves the limited memory bandwidth of mobile devices. The primary advantage of $\mathbf{C}_4^n$ Compact over CSR is its ability to store column indices more compactly. This efficiency comes from recognizing that multiple rows may share identical column indices after applying $\mathbf{C}_4^n$ kernels for pruning. To accomplish this, it utilizes three arrays: *Ro_Idx*, *Rc_Idx*, and *K_Idx*, to represent the row and column index for each kernel after reordering, and pattern style, respectively. As we don't need to store the column/row information for each non-zero elements, this compact storage saves significant memory usage ranging from 25% to 75% (the number depends on the pruning ratio in the sparse weights) compared with the CSR format.

**Layout Transformation Optimization** Our previous optimizations are dedicated for the sparse model, targeting *computational intensive operators*, e.g., Conv, MatMul. However, Mamba's architecture (computational graph), which relies heavily on *layout transformation operators* for tensor reshaping and transposing, demands high bandwidth memory (Gu & Dao, 2023a). This issue is par-

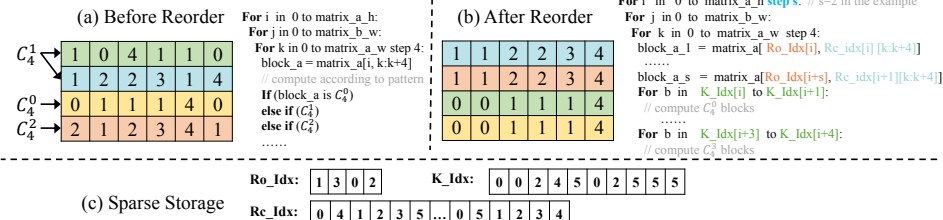

Figure 2: Weight reorder and sparse storage. Numbers in (a) and (b) represent the pattern styles. Three extra indexes will be utilized during execution. $Ro\_Idx$ represents the original row index after reordering, while $Rc\_Idx$ denotes the original column index for each reordered kernel. And $K\_Idx$ represents the stride information for kernels with identical patterns.

ticularly critical on mobile devices, where bandwidth is limited compared to desktop GPUs (Huynh et al., 2017; Lee et al., 2019), leading to significant overhead. To further improve the performance, we propose a layout transformation elimination strategy to fully eliminate the layout transformation operators while maintain the same accuracy. The key insight is that requiring the producer to create a layout based on the consumer's reduction dimension results in relatively low additional overhead compared to other options. First, our compiler extract the operator information (e.g., operator types, input feature sizes) and select the optimal layouts for each individual operator. Then, we eliminate the layout transformation operator in the computational graph (i.e., *Transpose* and *Reshape*). Finally, we align the data layout produced by one operator to the needs of the subsequent operator during computation, and the layout transformation elimination technique minimizes the overhead associated with explicit data transformations. The detailed formalized algorithm for layout transformation elimination is provided in Appendix B.2. This approach significantly benefits mobile devices with limited bandwidth by enhancing data locality and reducing memory access costs, as reflected in Table 3 (our dense vs. llama.cpp).

## 6 EXPERIMENTS

### 6.1 EXPERIMENT SETUP

**Sparse learning recipe** We use Mamba models to test the effectiveness of our method. Our approach covers a variety of Mamba models, with parameters ranging from 130M to 2.8B. The selective SSM architecture in Mamba primarily uses linear projections for both input and output in each block, with a significantly smaller number of SSM parameters (projections for $\Delta$, $\mathbf{B}$, $\mathbf{C}$, and $\mathbf{A}$). We apply sparsity to all these parameters. For sparse learning, we train over 1000 steps using the SGD optimizer at a starting learning rate of 0.1, decaying it by 0.1 every 200 steps. The original model weights are frozen and sparse learning for Mamba-2.8B takes 50 mins on a A6000 GPU. We generate optimal sparse mask for each layer with sparsity-oriented sparse learning independently. The results generated with latency-oriented sparse learning are included in Appendix C. We prune the model in one-shot and evaluate the task performance on multiple common sense reasoning datasets including LAMBADA (Paperno et al., 2016), HellaSwag (Zellers et al., 2019), PIQA (Bisk et al., 2020), Arc-easy (Clark et al., 2018), Arc-challenge (Clark et al., 2018), and WinoGrade (Sakaguchi et al., 2021). Perplexity on LAMBADA dataset and average accuracy on all mentioned datasets are provided.

**Testing bed** The latency evaluations are conducted on a Oneplus 11 mobile device equipped with Snapdragon 8 Gen 2 SoC, featuring an octa-core Kryo CPU and Qualcomm Adreno 740 GPU with 16GB memory. Tests utilize all threads on mobile CPUs and all pipelines on mobile GPUs. We report the performance comparison against llama.cpp (contributors, 2023a), as it is the only open-source framework to support Mamba on mobile devices yet. The GPU runs use 16-bit floating points while the CPU runs use 32-bit floating points. The data types used in the evaluation are aligned with llama.cpp's support for Mamba. Each experiment is repeated 50 times. Due to small variance, only average results are reported. The batch size is set to 1 for all models unless otherwise specified.

### 6.2 MAIN RESULTS

We show our main results of zero-shot performance with Mamba models with 130M, 370M, 790M, 1.4B, and 2.8B in Table 2. We compare our method against other semi-structure pruning methods, including SparseGPT (Frantar & Alistarh, 2023) and Wanda (Sun et al., 2023), with 2:4 and 4:8 patterns which can accelerate the sparse networks (Mishra et al., 2021). As observed, our proposed

Table 2: Main results of zero-shot performance with all scales of Mamba models. We compare against semi-structure pruning methods, including SparseGPT and Wanda, with 2:4 pattern.

| Method | Sparsity Ratio | LAMBDA PPL↓ | LAMBDA Acc↑ | HellaSwag Acc↑ | PIQA Acc↑ | Arc-E Acc↑ | Arc-C Acc↑ | WinoGrade Acc↑ | Avg. Acc↑ |
|---|---|---|---|---|---|---|---|---|---|
| Mamba-130M | \ | 16.07 | 44.3 | 35.3 | 64.5 | 48.0 | 24.3 | 51.9 | 44.7 |
| SparseGPT 2:4 | 50% | 69.80 | 26.1 | 29.8 | 58.8 | 37.5 | 22.7 | 52.4 | 37.9 |
| Wanda 2:4 | 50% | 212.91 | 15.4 | 29.4 | 56.8 | 38.7 | 21.1 | 52.5 | 35.6 |
| Ours | 30% | 17.47 | 42.1 | 34.5 | 64.0 | 47.1 | 23.8 | 53.6 | 44.2 |
| Ours | 50% | 28.97 | 35.2 | 32.2 | 60.8 | 41.6 | 24.2 | 51.9 | 41.0 |
| Ours | 70% | 57.29 | 27.7 | 29.9 | 58.9 | 36.4 | 22.9 | 50.1 | 37.8 |
| Mamba-370M | \ | 8.14 | 55.6 | 46.5 | 69.5 | 55.1 | 28.0 | 55.3 | 51.6 |
| SparseGPT 2:4 | 50% | 27.93 | 35.8 | 34.7 | 61.5 | 40.6 | 23.1 | 51.8 | 41.2 |
| Wanda 2:4 | 50% | 82.52 | 22.6 | 31.9 | 60.1 | 40.3 | 22.4 | 51.7 | 38.1 |
| Ours | 30% | 8.55 | 54.6 | 44.9 | 68.8 | 52.7 | 27.5 | 55.3 | 50.6 |
| Ours | 50% | 12.33 | 47.9 | 40.2 | 64.7 | 47.6 | 26.5 | 54.3 | 46.9 |
| Ours | 70% | 22.09 | 38.9 | 35.2 | 61.4 | 40.5 | 23.7 | 51.6 | 41.9 |
| Mamba-790M | \ | 6.02 | 62.7 | 55.1 | 72.1 | 61.2 | 29.5 | 56.1 | 57.1 |
| SparseGPT 2:4 | 50% | 13.69 | 46.2 | 40.1 | 64.0 | 45.0 | 24.7 | 55.3 | 45.9 |
| Wanda 2:4 | 50% | 43.76 | 28.5 | 36.9 | 62.8 | 43.3 | 22.6 | 55.3 | 41.5 |
| Ours | 30% | 6.21 | 60.7 | 53.6 | 71.8 | 58.7 | 28.4 | 56.2 | 54.9 |
| Ours | 50% | 7.87 | 56.0 | 48.0 | 68.9 | 51.6 | 26.3 | 55.9 | 51.1 |
| Ours | 70% | 11.85 | 48.9 | 41.8 | 64.7 | 44.5 | 25.8 | 55.4 | 46.9 |
| Mamba-1.4B | \ | 5.04 | 64.9 | 59.1 | 74.2 | 65.5 | 32.8 | 61.5 | 59.7 |
| SparseGPT 2:4 | 50% | 8.87 | 54.3 | 44.5 | 66.5 | 49.1 | 24.3 | 54.5 | 48.9 |
| Wanda 2:4 | 50% | 32.72 | 31.6 | 38.2 | 63.9 | 46.8 | 22.8 | 53.7 | 42.8 |
| Ours | 30% | 5.08 | 65.1 | 58.2 | 73.2 | 64.0 | 32.4 | 60.7 | 58.9 |
| Ours | 50% | 5.65 | 62.5 | 52.7 | 70.7 | 58.6 | 29.0 | 59.0 | 55.4 |
| Ours | 60% | 7.30 | 57.5 | 46.0 | 68.3 | 51.6 | 26.3 | 57.4 | 51.2 |
| Ours | 70% | 17.40 | 43.2 | 31.7 | 62.5 | 43.4 | 19.5 | 55.1 | 42.6 |
| Ours | 75% | 19.65 | 42.0 | 35.7 | 61.1 | 41.2 | 22.9 | 54.4 | 42.9 |
| Mamba-2.8B | \ | 4.23 | 69.2 | 66.1 | 75.2 | 69.7 | 36.3 | 63.5 | 63.3 |
| SparseGPT 2:4 | 50% | 5.11 | 65.6 | 52.1 | 70.0 | 56.0 | 27.6 | 59.8 | 55.2 |
| Wanda 2:4 | 50% | 10.49 | 50.0 | 48.0 | 65.8 | 54.9 | 26.3 | 56.2 | 50.2 |
| Ours | 30% | 4.18 | 69.4 | 65.2 | 75.6 | 69.2 | 36.4 | 62.6 | 63.1 |
| Ours | 50% | 4.26 | 68.9 | 60.2 | 72.6 | 65.2 | 31.5 | 61.1 | 59.9 |
| Ours | 60% | 4.72 | 66.7 | 54.0 | 71.1 | 57.3 | 28.4 | 59.4 | 56.1 |
| Ours | 70% | 7.51 | 58.8 | 43.3 | 64.6 | 46.6 | 25.2 | 58.3 | 49.5 |
| Ours | 75% | 15.86 | 46.6 | 36.2 | 61.0 | 40.7 | 22.8 | 56.2 | 43.9 |

pruning method shows a consistently significant improvement over SparseGPT and Wanda at the same sparsity ratio in terms of the average accuracy across the six datasets. Taking Mamba-1.4B as an example, our method with a sparsity ratio of 50% increases the average accuracy by at least 3% compared with all baseline methods. Besides, compared to other methods with fixed 2:4 or 4:8 patterns, our method can achieve flexible sparsity ratios, such as a 30% sparsity ratio with an accuracy comparable to the dense model. Full results are provided in Appendix D. In summary, the results achieved by our method indicate the effectiveness of our pruning technique in maintaining performance across different model scales and tasks. We also provide the comparison to transformer-based models in Table A3 and Table A4, and we achieve superior task performance in terms of perplexity and average accuracy compared to transformer-based models under the same model size.

## 6.3 LATENCY RESULTS

We conduct the latency measurements of our method under different model scales on both the CPU and GPU of the Snapdragon 8 Gen 2 mobile platform to show the actual acceleration performance. The generation latency results are shown in Table 3. Our results demonstrate that, compared with the dense model, our method achieves approximately $1.1\times$ to $1.2\times$ acceleration at 30% sparsity, $1.3\times$ to $1.5\times$ acceleration at 50% sparsity and nearly $1.6\times$ to $1.7\times$ acceleration at 70% and 75% sparsity, on both mobile CPU and GPU. To demonstrate the performance gains from our general optimizations and layout transformation elimination, our compiler (dense) achieves up-to $4.5\times$ speedup compared to llama.cpp (dense since llama.cpp does not yet support sparse models) on mobile CPU. It is also

Table 3: Latency results of Mamba with different scales and 64 sequence length. SPD stands for speedup over llama.cpp (red) and our dense baseline (blue).

| Mamba | Method | Sparsity | Mobile CPU | | Mobile GPU | |
|---|---|---|---|---|---|---|
| | | | Token/s | SPD | Token/s | SPD |
| 130M | llama.cpp | 0% | 3.5 | 1.0× | - | - |
| | ours | 0% | 11.2 | 3.2×/1.0× | 33.6 | 1.0× |
| | ours | 30% | 12.1 | 3.5×/1.1× | 40.7 | 1.2× |
| | ours | 50% | 14.9 | 4.3×/1.3× | 50.4 | 1.5× |
| | ours | 70% | 18.1 | 5.2×/1.6× | 54.8 | 1.6× |
| 370M | llama.cpp | 0% | 1.5 | 1.0× | - | - |
| | ours | 0% | 6.1 | 4.1×/1.0× | 19.6 | 1.0× |
| | ours | 30% | 6.9 | 4.6×/1.1× | 21.5 | 1.1× |
| | ours | 50% | 7.9 | 5.5×/1.3× | 25.4 | 1.3× |
| | ours | 70% | 9.8 | 6.5×/1.6× | 31.3 | 1.6× |
| 790M | llama.cpp | 0% | 0.7 | 1.0× | - | - |
| | ours | 0% | 3.1 | 4.5×/1.0× | 12.8 | 1.0× |
| | ours | 30% | 3.3 | 4.7×/1.1× | 14.1 | 1.1× |
| | ours | 50% | 4.3 | 6.3×/1.4× | 16.6 | 1.3× |
| | ours | 70% | 5.0 | 7.0×/1.6× | 20.4 | 1.6× |
| 1.4B | llama.cpp | 0% | 0.6 | 1.0× | - | - |
| | ours | 0% | 2.1 | 3.5×/1.0× | 10.1 | 1.0× |
| | ours | 30% | 2.5 | 4.1×/1.2× | 10.9 | 1.1× |
| | ours | 50% | 2.9 | 4.8×/1.4× | 13.1 | 1.3× |
| | ours | 75% | 3.3 | 5.8×/1.6× | 17.2 | 1.7× |
| 2.8B | llama.cpp | 0% | 0.4 | 1.0× | - | - |
| | ours | 0% | 1.9 | 4.0×/1.0× | 7.7 | 1.0× |
| | ours | 30% | 2.2 | 4.7×/1.2× | 9.1 | 1.2× |
| | ours | 50% | 2.6 | 5.5×/1.4× | 11.5 | 1.5× |
| | ours | 75% | 3.0 | 6.5×/1.6× | 13.0 | 1.7× |

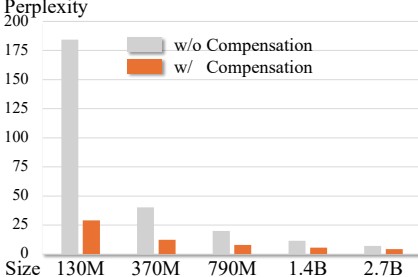

Table 4: Ablation analysis with or without weight compensation.

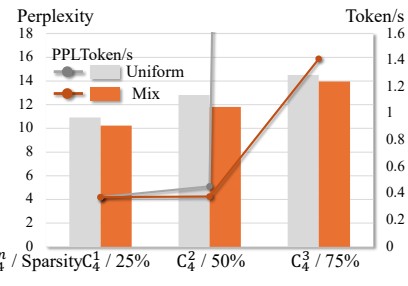

Table 5: Ablation analysis of uniform and mixed kernels strategies.

worth pointing out that our compiler works efficiently on mobile GPU, achieving approximately $3\times$ to $5\times$ speedup compared our CPU version. GPU results are not included for llama.cpp as it is not yet supported. Meanwhile, we present other hardware results including the peak memory during inference and energy consumption in Table A5 and Table A6, respectively. We observe that our sparse models require only a slight increase in memory while achieving notable energy savings and significant inference acceleration, demonstrating the efficiency of our method. Additionally, the results tested on another edge device, Xiaomi 6, which is equipped with a Snapdragon 835, featuring an octa-core CPU and an Adreno GPU with 6GB of memory, are included in Table A7.

### 6.4 ABLATION STUDY

We conduct the ablation study in terms of weight compensation, and the results are shown in Figure 4. We experiment with two settings: with or without weight compensation, for each scale of the Mamba model on the LAMBDA dataset. The consistent better task performance with our weight compensation technique demonstrates its effectiveness.

To verify the effectiveness of sparse learning, we also employ a uniform kernel strategy for the ablation as shown in Figure 5. The perplexity results are evaluated with Mamba-2.8B model on the LAMBADA dataset, and the latency results are tested on mobile CPU. Compared with the uniform kernel strategy, our mixed kernel strategy with sparse learning achieves better task performance while maintaining similar latency performance, especially when the sparsity exceeds 25%.

### 7 CONCLUSION

In this paper, we propose a sparse learning framework with special compiler kernels for the acceleration of Mamba models on mobile devices. We introduce the $\mathbf{C}_4^n$ kernel designed to prune $n$ elements from every four contiguous weights. Then, we propose the sparse learning framework to explore the optimal pruning strategy with kernels for the weights in Mamba models. Besides, we propose the weight compensation method to rectify the weights in sparse models with the pruned weights. Finally, we implement the sparse strategy with kernels on mobile devices with the $\mathbf{C}_4^n$-specific optimizations and the layout transformation elimination strategy. Experiments show the effectiveness of our method for the acceleration of Mamba models on mobile devices.

ACKNOWLEDGMENT

We sincerely appreciate the contributions of everyone involved in this work, with special thanks to the anonymous reviewers for their valuable feedback. This research was partially supported by the National Science Foundation (NSF) under awards CCF-2428108 and OAC-2403090. Any errors and opinions are not those of the NSF and are attributable solely to the author(s).

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
