# Appendix

## A    PROOF OF THEOREM 4.1

We first reformulate Problem (5) as there are two variables $\widehat{\mathbf{W}}$ and $\mathbf{M}$, leading to difficulties for optimization. For each linear layer, we try to minimize the difference of the linear outputs (measured by $\ell_2$ norm) before and after pruning, i.e., $\|(\mathbf{W}+\delta\mathbf{W})\mathbf{X}-\mathbf{W}\mathbf{X}\|_2^2 = \|\delta\mathbf{W}\mathbf{X}\|_2^2$. A mask $\mathbf{M}$ denotes the pruned locations, i.e., the values indexed $(q_i, p_{i1}), (q_i, p_{i2}), ..., (q_i, p_{ik_i}), \forall i \in \{1, ..., k\}$ in $\mathbf{M}$ are all zeros. $(q_i, p_{ij})$ denotes the pruned weight index in the $q_i^{th}$ row and $p_{ij}^{th}$ column. The pruned weights are distributed in $k$ rows and there are $k_i$ pruned elements in the $i^{th}$ row with pruned elements. To make the problem tractable, the pruned weights $(q_i, p_{ij})$ are set to zero, i.e. $[\mathbf{W} + \delta\mathbf{W}]_{q_i, p_{ij}} = 0$. To minimize the loss incurred by pruning, the other unpruned weights are optimized for minimal loss. Problem (5) can be reformulated as following,

$$\min_{\delta\mathbf{W}} \ \mathcal{L}(\delta\mathbf{W}) = \|\delta\mathbf{W}\mathbf{X}\|^2,$$

$$s.t. \ \mathbf{e}_{q_i}^T \delta\mathbf{W}\mathbf{e}_{p_{i1}} + [W]_{q_i, p_{i1}} = 0,$$

$$\mathbf{e}_{q_i}^T \delta\mathbf{W}\mathbf{e}_{p_{i2}} + [W]_{q_i, p_{i2}} = 0,$$

$$......$$

$$\mathbf{e}_{q_i}^T \delta\mathbf{W}\mathbf{e}_{p_{ik_i}} + [W]_{q_i, p_{ik_i}} = 0,$$

$$\forall i \in \{1, ..., k\}, \tag{15}$$

where $\mathbf{e}_{q_i}$ is a one-hot vector with the $q_i^{th}$ element as 1 and all others as 0, and $\mathbf{e}_{p_{ij}}$ has similar meanings. $\mathbf{e}_{q_i}^T \delta\mathbf{W}\mathbf{e}_{p_{ij}}$ denotes the weight in the $q_i^{th}$ row and the $p_{ij}^{th}$ column of $\delta\mathbf{W}$. It is equivalent to Problem (5).

It can be transformed to vector representation,

$$\min_{\delta\mathbf{W}} \ \mathcal{L}(\delta\mathbf{W}) = \|\delta\mathbf{W}\mathbf{X}\|^2,$$

$$s.t. \ \mathbf{e}_{q_1}^T \delta\mathbf{W}\mathbf{M}_1 + \mathbf{W}_{q_1} = \mathbf{0},$$

$$\mathbf{e}_{q_2}^T \delta\mathbf{W}\mathbf{M}_2 + \mathbf{W}_{q_2} = \mathbf{0},$$

$$......$$

$$\mathbf{e}_{q_k}^T \delta\mathbf{W}\mathbf{M}_k + \mathbf{W}_{q_k} = \mathbf{0}, \tag{16}$$

where $\mathbf{M}_i \in \mathbb{R}^{m \times k_i}$ with $[M_1]_{:,j} = \mathbf{e}_{p_{ij}}$, and $\mathbf{W}_{qi} = [[\mathbf{W}]_{q_i, p_{i1}}, [\mathbf{W}]_{q_i, p_{i2}}, ..., [\mathbf{W}]_{q_i, p_{ik_i}}] \in \mathbb{R}^{1 \times k_i}$. $M_i$ is a collection of all pruned column indexes in the $q_i^{th}$ row, and $\mathbf{W}_{qi}$ is a collection of all pruned weight values in the $q_i^{th}$ row. $\mathbf{W}_{qi}$ can be further represented as $\mathbf{e}_{q_i}^T \mathbf{W}\mathbf{M_i}$.

The Lagrange function for Problem (16) is

$$\mathscr{L}(\delta\mathbf{W}, \lambda) = \|\delta\mathbf{W}\mathbf{X}\|^2 + (\mathbf{e}_{q_1}^T \delta\mathbf{W}\mathbf{M}_1 + \mathbf{W}_{q_1})\lambda_1 + (\mathbf{e}_{q_2}^T \delta\mathbf{W}\mathbf{M}_2 + \mathbf{W}_{q_2})\lambda_2$$

$$+ ... + (\mathbf{e}_{q_k}^T \delta\mathbf{W}\mathbf{M}_k + \mathbf{W}_{q_k})\lambda_k,$$

$$= \text{Tr}(\mathbf{X}^T \delta\mathbf{W}^T \delta\mathbf{W}\mathbf{X}) + \sum_i (\mathbf{e}_{q_i}^T \delta\mathbf{w}\mathbf{M}_i + \mathbf{W}_{q_i})\lambda_i, \tag{17}$$

where $\lambda_i \in R^{k_i \times 1}$ denotes the Lagrange multiplier corresponding to the constraint for the $q_i^{th}$ row in Problem (16). $\lambda_i = [\lambda_{i1}, \lambda_{i2}, ......, \lambda_{ik_i}]$ and each $\lambda_{ij}$ corresponds to the constraint $\mathbf{e}_{q_i}^T \delta\mathbf{W}\mathbf{e}_{p_{ij}} + [\mathbf{W}]_{q_i, p_{ij}} = 0$ in Problem (15). The trace function $\text{Tr}(\cdot)$ computes the $\ell_2$ norm of $\delta\mathbf{W}\mathbf{X}$.

The gradients with reference to $\delta\mathbf{W}$ should be 0.

$$\frac{\delta\mathscr{L}(\delta\mathbf{W}, \lambda)}{\delta(\delta\mathbf{W})} = 2\delta\mathbf{W}\mathbf{X}\mathbf{X}^T + \sum_i \mathbf{e}_{q_i}\lambda_i^T \mathbf{M}_i^T = 0. \tag{18}$$

We can obtain $\delta\mathbf{W}$ as below,

$$\delta\mathbf{W} = -\left(\sum_i \mathbf{e}_{q_i}\lambda_i^T \mathbf{M}_i^T\right)(2\mathbf{X}\mathbf{X}^T)^{-1}. \tag{19}$$

By applying Equation (19) in Equation (17), we have the following,

$$
\begin{aligned}
g(\lambda) =&\ \mathrm{Tr}\left(\mathbf{X}^T(2\mathbf{X}\mathbf{X}^T)^{-1}\left(\sum_i \mathbf{M}_i\lambda_i\mathbf{e}_{q_i}^T\right)\left(\sum_i \mathbf{e}_{q_i}\lambda_i^T\mathbf{M}_i^T\right)(2\mathbf{X}\mathbf{X}^T)^{-1}\mathbf{X}\right) \\
&- \sum_i \mathbf{e}_{q_i}^T\left(\sum_i \mathbf{e}_{q_i}\lambda_i^T\mathbf{M}_i^T\right)(2\mathbf{X}\mathbf{X}^T)^{-1}\mathbf{M}_i\lambda_i + \sum_i \mathbf{W}_{q_i}\lambda_i \\
=&\ \mathrm{Tr}\left(\mathbf{X}^T(2\mathbf{X}\mathbf{X}^T)^{-1}\left(\sum_i \mathbf{M}_i\lambda_i\lambda_i^T\mathbf{M}_i^T\right)(2\mathbf{X}\mathbf{X}^T)^{-1}\mathbf{X}\right) \\
&- \sum_i \lambda_i^T\mathbf{M}_i^T(2\mathbf{X}\mathbf{X}^T)^{-1}\mathbf{M}_i\lambda_i + \sum_i \mathbf{W}_{q_i}\lambda_i \\
=&-\frac{1}{2}\sum_i \lambda_i^T\mathbf{M}_i^T(2\mathbf{X}\mathbf{X}^T)^{-1}\mathbf{M}_i\lambda_i + \sum_i \mathbf{W}_{q_i}\lambda_i
\end{aligned}
\tag{20}
$$

Note that $\mathbf{e}_{q_i}^T\mathbf{e}_{q_i} = 1$ and $\mathbf{e}_{q_i}^T\mathbf{e}_{q_j} = 0$, if $i \neq js$. Besides, we can switch the position of $\mathbf{X}^T(2\mathbf{X}\mathbf{X}^T)^{-1}\mathbf{M}_i\lambda_i$ and $\lambda_i^T\mathbf{M}_i^T(2\mathbf{X}\mathbf{X}^T)^{-1}\mathbf{X}$ in the trace function.

The gradients with reference to $\lambda$ should be 0.

$$
\frac{\delta g(\lambda)}{\delta\lambda_i} = -\mathbf{M}_i^T(2\mathbf{X}\mathbf{X}^T)^{-1}\mathbf{M}_i\lambda_i + \mathbf{W}_{q_i}^T = \mathbf{0}, \forall i.
\tag{21}
$$

We can obtain the optimal $\lambda$ as below,

$$
\lambda_i^* = [\mathbf{M}_i^T(2\mathbf{X}\mathbf{X}^T)^{-1}\mathbf{M}_i]^{-1}\mathbf{W}_{q_i}^T, \forall i.
\tag{22}
$$

The optimal $\delta\mathbf{W}$ can be derived as below,

$$
\begin{aligned}
\delta\mathbf{W}^* =&-\left(\sum_i \mathbf{e}_{q_i}\mathbf{W}_{q_i}[\mathbf{M}_i^T(2\mathbf{X}\mathbf{X}^T)^{-1}\mathbf{M}_i]^{-1}\mathbf{M}_i^T\right)\times(2\mathbf{X}\mathbf{X}^T)^{-1} \\
=&-\left(\sum_i \mathbf{e}_{q_i}\mathbf{e}_{q_i}^T\mathbf{W}\mathbf{M}_i[\mathbf{M}_i^T(2\mathbf{X}\mathbf{X}^T)^{-1}\mathbf{M}_i]^{-1}\mathbf{M}_i^T\right)\times(2\mathbf{X}\mathbf{X}^T)^{-1}.
\end{aligned}
\tag{23}
$$

The minimal loss/error corresponding to the optimal $\delta\mathbf{W}$ can be obtained by

$$
\begin{aligned}
L^* =&\frac{1}{2}\sum_i \lambda_i^T\mathbf{M}_i^T(2\mathbf{X}\mathbf{X}^T)^{-1}\mathbf{M}_i\lambda_i \\
=&\frac{1}{2}\sum_i \mathbf{W}_{q_i}[\mathbf{M}_i^T(2\mathbf{X}\mathbf{X}^T)^{-1}\mathbf{M}_i]^{-1}\mathbf{W}_{q_i}^T \\
=&\frac{1}{2}\sum_i \mathbf{e}_{q_i}^T\mathbf{W}\mathbf{M}_i[\mathbf{M}_i^T(2\mathbf{X}\mathbf{X}^T)^{-1}\mathbf{M}_i]^{-1}\mathbf{M}_i^T\mathbf{W}^T\mathbf{e}_{q_i} \\
=&\frac{1}{2}\sum_i [\mathbf{W}\mathbf{M}_i[\mathbf{M}_i^T(2\mathbf{X}\mathbf{X}^T)^{-1}\mathbf{M}_i]^{-1}\mathbf{M}_i^T\mathbf{W}^T]_{q_i,q_i}.
\end{aligned}
\tag{24}
$$

# B COMPILER OPTIMIZATIONS

This section further details our compiler optimizations, including operator fusion and tensor enhancements. Next, we describe our formalized algorithm for eliminating layout transformations.

## B.1 GENERAL OPTIMIZATIONS

**Operator Fusion** The Mamba model comprises a series of diverse operators, such as *Conv* and *MatMul*. This results in frequent data movement between operators, presenting significant challenges for memory throughput demands. With more than 100 different types of operators, existing

frameworks like TFLite, Pytorch-Mobile, and llama.cpp rely on a fixed pattern matching strategy to identify and fuse operator combinations. However, this approach fails to recognize new operator combinations due to the vast potential combination space among the operators. To address this issue, we have developed an advanced operator fusion strategy. First, we categorize operators into groups based on their input-output mapping relationships—One-to-One, One-to-Many, Many-to-Many—to assess the feasibility of fusion. Next, our loop fusion framework operates at a high-level operator abstraction with only three categories. This simplification significantly broadens fusion possibilities by making it easier to classify individual operators and their combinations. Compared to traditional operator fusion supported by frameworks like llama.cpp,TFLite,and Pytorch-Mobile our method offers greater flexibility and more opportunities for aggressive optimization.

**Tensor Optimizations** We also incorporate a suite of tensor optimizations such as memory planning, constant folding, and shape inference into our framework, similar to those found in other frameworks. What sets our tensor optimizations apart is the introduction of graph rewriting rules that utilize mathematical properties to optimize Mamba computations. This approach not only lowers evaluation costs but also simplifies operator fusion.

## B.2 ALGORITHM FOR LAYOUT TRANSFORMATION ELIMINATION

List 1 presents our algorithm for selecting and eliminating layouts. We determine the actions for each edge (connecting a producer and a consumer) using a depth-first search, starting from the graph's inputs (Line 3 to Line 4). The action consists of eliminating (eliminate the layout transformation), fusing, as well as searching an optimal data layout for an operator. After collecting the actions for all edges, we focus on processing operator fusion and layout mapping. We extend the fusion algorithm in Section B.1 while employing different strategies when one edge has multiple consumers (Line 36). When an operator can be fused with multiple predecessors, we fuse it with the operator that has maximum intermediate results to minimize index computation overhead. If an output tensor has multiple consumers and those consumers have less than 3 reduction dimensions (the dimension performs reduction operation during the computation), we use a layout to align these dimensions iteratively (Line 40 to Line 42). However, if there are more reduction dimensions, we maintain several copies of data with different layouts, and each layout is in this optimized combined format (Line 45 to Line 47).

```
1  ecg = build_ecg(cg)    # Build graph with CD info
2  # analysis actions in ecg
3  for edge in ecg.in_edges:
4    depth_first_search(edge, ecg, True)
5  # process search/fuse/eliminate actions in ecg
6  for edge in ecg.in_edges:
7    depth_first_search(edge, ecg, False)
8
9  def depth_first_process(in_edge, ecg, analysis):
10   out_edges = in_edge.consumer.out_edges
11   # Recurse for its consumer's output edges
12   for out_edge in out_edges:
13     if analysis:
14       determine_action(in_edge, out_edges, ecg)
15     else:
16       process_action(in_edge, out_edges, ecg)
17     depth_first_process(out_edge, ecg, analysis)
18
19  def determine_action(curr_edge, next_edge, ecg):
20    # Check consumer types of curr and next edges
21    t1, t2 = curr_edge.ctype, next_edge.ctype
22    if t1 == t2 == 'ILD&Var':
23      curr_edge.search = True
24      next_edge.search = True
25    elif t1 == 'ILD&Var' t2 == 'ILI&Var':
26      curr_edge.search = True
27      next_edge.fuse = True
28    elif t1 == 'ILD&Var' t2 == 'ILI&Fixed':
29      curr_edge.search = True
```

```
30      eliminate_edge(next_edge, ecg)
31    # Continue for other combinations ...
32
33 def process_action(curr_edge, next_edge, ecg):
34    if curr_edge.fuse or next_edge.fuse:
35      # Fuse edge with its optimal consumer
36      try_fuse(curr_edge, next_edge, ecg)
37    # Check all successor edges to select layouts
38    out_edges = curr_edge.consumer.out_edges
39    cds = get_reduction_dims(out_edges)
40    if len(unique(cds)) < 3:
41      next_edge = merge_edges(out_edges)
42      set_physical_layout(curr_edge, next_edge)
43    else:
44      # Select 2 edges to merge and remaining
45      merged, rem = select_edges(out_edges, cfg)
46      set_physical_layout(curr_edge, merged)
47      insert_implicit_convert(rem)
48
49 def set_physical_layout(curr_edge, out_edge):
50    # Map reduction dims to consecutive 2.5D memory
```

Listing 1: Algorithm for Layout Transformation Elimination

## C    LATENCY-ORIENTED SPARSE LEARNING

We provide the latency-oriented sparse learning results on the Mamba-2.8B model in Table A1. It is observed that to achieve faster inference acceleration, the task performance of sparse models tends to decrease compared to accuracy-oriented sparse learning.

## D    FULL RESULTS

We provide the full sparse results with sparsity ratios vary from 10% to 80% and additional 4:8 sparse patterns for SparseGPT and Wanda methods in Table A2. We can find that our method can achieve the better performance with larger sparsity than SparseGPT and Wanda with 2:4 or 4:8 patterns in all scales of the Mamba models, which verifies the effectiveness of our sparse learning framework.

## E    MEMORY AND ENERGY CONSUMPTION COMPARISON

## F    PORTABILITY EVALUATION

## G    COMPARE WITH TRANSFORMERS

We compare the 50% sparsity model generated by our method with various transformer-based models, as shown in Table A3. Our method consistently outperforms most other transformer-based models across different scales. Especially, with 50% sparsity and 2.8B model size, our method can achieve similar task performance as OPT-6.7B model, when enjoying the inference acceleration on mobile devices.

## H    LLaMA RESULTS

We deliver the perplexity results of LLaMA family on WikiText2 with 50% sparsity compared to the SparseGPT Frantar & Alistarh (2023) and Wanda Sun et al. (2023) in Table A4. Our method achieves better performance than other two methods.

## I ABLATION FOR DAMPENING RATIO

We conducted an ablation study on the dampening ratio to evaluate its impact on model performance, using Mamba-2.8B at 50% sparsity on LAMBDA dataset, as illustrated in Figure A1. The results indicate that as the dampening ratio increases, the model performance progressively deteriorates.

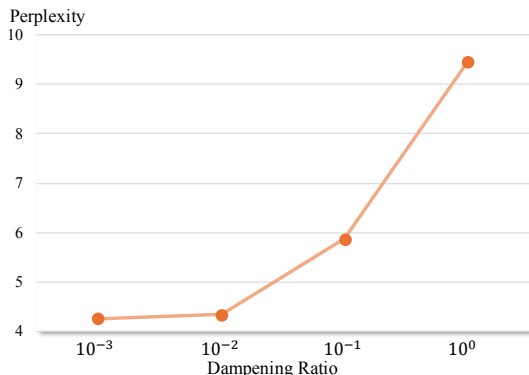

Figure A1: Ablation for the dampening ratio.

Table A1: Latency-oriented sparse learning results of Mamba-2.8B model.

| Mamba | Sparse Learning | Mobile CPU Token/s | LAMBADA PPL ↓ | LAMBADA Acc ↑ | HellaSwag Acc ↑ | PIQA Acc ↑ | Arc-E Acc ↑ | Arc-C Acc ↑ | WinoGrade Acc ↑ | Avg. Acc ↑ |
|---|---|---|---|---|---|---|---|---|---|---|
| 1.4B | Baseline | 2.06 | 5.04 | 64.9 | 59.1 | 74.2 | 65.5 | 32.8 | 61.5 | 59.7 |
| | ✓ | 3.64 | 7.01 | 58.63 | 37.36 | 66.64 | 52.35 | 22.38 | 57.22 | 49.10 |
| | ✓ | 4.31 | 39.67 | 34.23 | 29.56 | 61.59 | 40.53 | 17.92 | 53.43 | 39.54 |
| 2.8B | Baseline | 1.85 | 4.23 | 69.2 | 66.1 | 75.2 | 69.7 | 36.3 | 63.5 | 63.3 |
| | ✓ | 2.54 | 6.61 | 60.57 | 38.16 | 68.23 | 52.86 | 23.55 | 59.04 | 50.40 |
| | ✓ | 3.04 | 20.28 | 43.2 | 32.25 | 58.13 | 39.89 | 19.28 | 54.06 | 41.14 |

## J PEAK MEMORY ON MOBILE

In Table A5, we present the peak memory usage of our framework on mobile devices. The Sparse column indicates the peak memory consumption for our framework. The sparse weight storage method maintains a consistent memory size regardless of the sparsity ratio. The sparse model requires only a slight increase in memory compared to the dense model, ranging from 2.4% to 5.5%, while delivering significantly faster inference. This demonstrates the efficiency and practicality of our approach.

## K ENERGY CONSUMPTION ON MOBILE

We present the energy consumption results of our framework with different model scales and sparsity ratios on mobile devices in Table A6. Compared to the dense model, our sparse model demonstrates a significant energy saving ratio, varying from 2.6% to 45.8%. The reduction in energy consumption is primarily attributed to our hardware design, which optimizes the proposed framework to significantly reduce computational demands. An important observation is that energy consumption decreases as the sparsity increases, highlighting the efficiency of our method in leveraging sparsity to minimize resource usage.

## L BENCHMARK ON OTHER EDGE DEVICES

We also conducted latency evaluations on a low-end device, the Xiaomi 6, which is equipped with a Snapdragon 835 featuring an octa-core CPU and an Adreno 540 GPU with 6GB of memory.

The results are presented in Table A7. Due to the memory limitations, we report the results for Mamba models with 130M, 370M, and 790M model scales. The results show similar trends: our dense version achieves a speedup ranging from $3.3\times$ to $4.4\times$ compared to llama.cpp on the mobile CPU. Meanwhile, compared to our dense version, our sparse method demonstrates considerable acceleration, achieving approximately $1.1\times$ to $1.2\times$ speedup at 30% sparsity, $1.1\times$ to $1.3\times$ speedup at 50% sparsity, and $1.2\times$ to $1.8\times$ speedup at 70% sparsity. These results demonstrate that our proposed method is both compatible and efficient across different edge devices.

Table A2: Full results for Mamba models with different scales.

| Method | Sparsity Ratio | LAMBADA PPL ↓ | LAMBADA Acc ↑ | HellaSwag Acc ↑ | PIQA Acc ↑ | Arc-E Acc ↑ | Arc-C Acc ↑ | WinoGrade Acc ↑ | Avg. Acc ↑ |
|---|---|---|---|---|---|---|---|---|---|
| Mamba-130M | \ | 16.07 | 44.3 | 35.3 | 64.5 | 48 | 24.3 | 51.9 | 44.71 |
| SparseGPT 2:4 | 50% | 69.80 | 26.08 | 29.76 | 58.81 | 37.54 | 22.70 | 52.41 | 37.88 |
| SparseGPT 4:8 | 50% | 43.96 | 31.24 | 30.83 | 59.74 | 39.52 | 22.35 | 51.30 | 39.16 |
| Wanda 2:4 | 50% | 212.91 | 15.35 | 29.38 | 56.80 | 38.72 | 21.08 | 52.49 | 35.64 |
| Wanda 4:8 | 50% | 90.46 | 22.61 | 30.65 | 58.00 | 39.69 | 22.70 | 52.33 | 37.66 |
| | 10% | 16.14 | 43.92 | 35.27 | 64.25 | 48.44 | 24.66 | 52.17 | 44.79 |
| | 20% | 16.35 | 43.74 | 35.03 | 64.69 | 47.98 | 24.57 | 51.07 | 44.51 |
| | 30% | 17.47 | 42.09 | 34.49 | 63.98 | 47.14 | 23.81 | 53.59 | 44.18 |
| Ours | 40% | 20.28 | 40.00 | 34.04 | 63.06 | 45.71 | 24.23 | 50.12 | 42.86 |
| | 50% | 28.97 | 35.20 | 32.21 | 60.83 | 41.58 | 24.23 | 51.93 | 41.00 |
| | 60% | 41.69 | 31.23 | 31.56 | 59.89 | 38.68 | 23.45 | 50.56 | 44.16 |
| | 70% | 57.29 | 27.67 | 29.90 | 58.87 | 36.41 | 22.87 | 50.12 | 37.64 |
| Mamba-370M | \ | 8.14 | 55.6 | 46.5 | 69.5 | 55.1 | 28 | 55.3 | 50.0 |
| SparseGPT 2:4 | 50% | 27.93 | 35.78 | 34.69 | 61.48 | 40.61 | 23.12 | 51.78 | 41.24 |
| SparseGPT 4:8 | 50% | 17.19 | 43.28 | 37.55 | 62.19 | 43.73 | 24.57 | 52.64 | 43.99 |
| Wanda 2:4 | 50% | 82.52 | 22.55 | 31.85 | 60.12 | 40.28 | 22.35 | 51.70 | 38.14 |
| Wanda 4:8 | 50% | 33.12 | 32.93 | 35.69 | 62.95 | 43.10 | 24.15 | 52.49 | 41.89 |
| | 10% | 8.13 | 55.73 | 46.49 | 69.53 | 54.59 | 27.90 | 55.88 | 51.69 |
| | 20% | 8.20 | 55.42 | 45.91 | 69.10 | 54.80 | 27.90 | 55.80 | 51.49 |
| | 30% | 8.55 | 54.57 | 44.88 | 68.82 | 52.69 | 27.47 | 55.33 | 50.63 |
| Ours | 40% | 9.59 | 52.42 | 43.33 | 68.34 | 50.55 | 27.56 | 54.30 | 49.42 |
| | 50% | 12.33 | 47.88 | 40.21 | 64.69 | 47.64 | 26.54 | 54.30 | 46.88 |
| | 60% | 16.78 | 43.21 | 37.68 | 62.56 | 43.43 | 25.12 | 52.95 | 48.79 |
| | 70% | 22.09 | 38.91 | 35.22 | 61.37 | 40.53 | 23.72 | 51.62 | 41.90 |
| Mamba-790M | \ | 6.02 | 62.7 | 55.1 | 72.1 | 61.2 | 29.5 | 56.1 | 57.1 |
| SparseGPT 2:4 | 50% | 13.69 | 46.24 | 40.07 | 63.98 | 44.95 | 24.74 | 55.33 | 45.89 |
| SparseGPT 4:8 | 50% | 9.88 | 51.10 | 43.94 | 66.49 | 47.94 | 24.49 | 55.09 | 48.18 |
| Wanda 2:4 | 50% | 43.76 | 28.45 | 36.88 | 62.79 | 43.27 | 22.61 | 55.25 | 41.54 |
| Wanda 4:8 | 50% | 19.73 | 39.90 | 41.81 | 65.34 | 47.98 | 24.57 | 53.99 | 45.60 |
| | 10% | 6.02 | 61.28 | 54.93 | 72.25 | 61.41 | 29.52 | 56.04 | 55.91 |
| | 20% | 6.10 | 61.42 | 54.67 | 71.71 | 60.14 | 28.92 | 55.96 | 55.47 |
| | 30% | 6.20 | 60.7 | 53.62 | 71.82 | 58.71 | 28.41 | 56.2 | 54.91 |
| Ours | 40% | 6.68 | 59.25 | 51.79 | 70.62 | 56.31 | 28.16 | 56.59 | 53.79 |
| | 50% | 7.87 | 56.01 | 47.96 | 68.88 | 51.56 | 26.28 | 55.88 | 51.10 |
| | 60% | 9.45 | 52.63 | 43.25 | 66.78 | 48.52 | 25.89 | 55.67 | 48.79 |
| | 70% | 11.85 | 48.85 | 41.78 | 64.74 | 44.53 | 25.77 | 55.41 | 46.85 |
| Mamba-1.4B | \ | 5.04 | 64.9 | 59.1 | 74.2 | 65.5 | 32.8 | 61.5 | 59.7 |
| SparseGPT 2:4 | 50% | 8.87 | 54.28 | 44.49 | 66.49 | 49.07 | 24.32 | 54.46 | 48.85 |
| SparseGPT 4:8 | 50% | 6.87 | 58.98 | 48.28 | 69.21 | 54.38 | 26.02 | 57.38 | 52.38 |
| Wanda 2:4 | 50% | 32.72 | 31.61 | 38.24 | 63.87 | 46.84 | 22.78 | 53.67 | 42.84 |
| Wanda 4:8 | 50% | 15.27 | 42.44 | 44.43 | 67.14 | 52.23 | 24.83 | 55.01 | 47.68 |
| | 10% | 5.04 | 64.99 | 59.03 | 74.05 | 65.15 | 32.68 | 60.62 | 59.42 |
| | 20% | 5.05 | 65.05 | 58.82 | 73.61 | 64.27 | 32.17 | 60.93 | 59.14 |
| | 30% | 5.08 | 65.05 | 58.24 | 73.23 | 64.02 | 32.42 | 60.69 | 58.94 |
| Ours | 40% | 5.21 | 64.27 | 56.27 | 72.63 | 61.36 | 31.14 | 59.98 | 57.61 |
| | 50% | 5.65 | 62.45 | 52.74 | 70.73 | 58.59 | 29.01 | 58.96 | 55.41 |
| | 60% | 7.30 | 57.52 | 46.04 | 68.28 | 51.64 | 26.28 | 57.38 | 51.19 |
| | 70% | 17.40 | 43.22 | 31.65 | 62.51 | 43.35 | 19.54 | 55.09 | 42.56 |
| | 75% | 19.65 | 41.96 | 35.74 | 61.10 | 41.16 | 22.87 | 54.38 | 42.87 |
| Mamba-2.8B | \ | 4.23 | 69.2 | 66.1 | 75.2 | 69.7 | 36.3 | 63.5 | 63.3 |
| SparseGPT 2:4 | 50% | 5.11 | 65.57 | 52.10 | 69.97 | 55.98 | 27.56 | 59.83 | 55.17 |
| SparseGPT 4:8 | 50% | 4.55 | 68.00 | 56.00 | 71.27 | 61.53 | 29.10 | 59.91 | 57.64 |
| Wanda 2:4 | 50% | 10.49 | 50.01 | 48.01 | 65.78 | 54.92 | 26.28 | 56.20 | 50.20 |
| Wanda 4:8 | 50% | 7.46 | 57.44 | 53.43 | 70.24 | 59.39 | 28.16 | 58.17 | 54.47 |
| | 10% | 4.22 | 69.11 | 66.02 | 75.24 | 69.82 | 36.52 | 63.38 | 63.35 |
| | 20% | 4.20 | 69.14 | 65.69 | 75.14 | 69.49 | 36.69 | 62.59 | 63.12 |
| | 30% | 4.18 | 69.42 | 65.17 | 75.57 | 69.23 | 36.43 | 62.59 | 63.07 |
| | 40% | 4.18 | 69.16 | 63.73 | 74.37 | 67.55 | 34.56 | 61.40 | 61.80 |
| Ours | 50% | 4.26 | 68.91 | 60.17 | 72.58 | 65.24 | 31.48 | 61.09 | 59.91 |
| | 60% | 4.72 | 66.74 | 53.95 | 71.06 | 57.32 | 28.41 | 59.35 | 56.14 |
| | 70% | 7.51 | 58.82 | 43.25 | 64.64 | 46.63 | 25.17 | 58.25 | 49.46 |
| | 75% | 15.86 | 46.59 | 36.16 | 61.04 | 40.70 | 22.78 | 56.20 | 43.91 |
| | 80% | 67.94 | 28.60 | 30.19 | 57.07 | 34.64 | 20.99 | 50.43 | 36.99 |

Table A3: Compare with transformer-based models with our 50% sparsity models.

| Method | LAMBADA | | HellaSwag | PIQA | Arc-E | Arc-C | WinoGrade | Avg. |
| | PPL ↓ | Acc ↑ | Acc ↑ | Acc ↑ | Acc ↑ | Acc ↑ | Acc ↑ | Acc ↑ |
|---|---|---|---|---|---|---|---|---|
| Hybrid H3-130M | 89.48 | 25.8 | 31.7 | 64.2 | 44.4 | 24.2 | 50.6 | 40.1 |
| Pythia-160M | 38.10 | 33.0 | 30.2 | 61.4 | 43.2 | 24.1 | 51.9 | 40.6 |
| Ours-130M | 28.97 | 35.2 | 32.2 | 60.8 | 41.6 | 24.2 | 51.9 | 41.0 |
| Hybrid H3-360M | 12.58 | 48.0 | 41.5 | 68.1 | 51.4 | 24.7 | 54.1 | 48.0 |
| Pythia-410M | 10.84 | 51.4 | 40.6 | 66.9 | 52.1 | 24.6 | 53.8 | 48.2 |
| Ours-370M | 12.33 | 47.9 | 40.2 | 64.7 | 47.6 | 26.5 | 54.3 | 46.9 |
| Pythia-1B | 7.92 | 56.1 | 47.2 | 70.7 | 57.0 | 27.1 | 53.5 | 51.9 |
| Ours-790M | 7.87 | 56.0 | 48.0 | 68.9 | 51.6 | 26.3 | 55.9 | 51.1 |
| GPT-Neo 1.3B | 7.50 | 57.2 | 48.9 | 71.1 | 56.2 | 25.9 | 54.9 | 52.4 |
| Hybrid H3-1.3B | 11.25 | 49.6 | 52.6 | 71.3 | 59.2 | 28.1 | 56.9 | 53.0 |
| OPT-1.3B | 6.64 | 58.0 | 53.7 | 72.4 | 56.7 | 29.6 | 59.5 | 55.0 |
| Pythia-14B | 6.08 | 61.7 | 52.1 | 71.0 | 60.5 | 28.5 | 57.2 | 55.2 |
| RWKV-1.5B | 7.04 | 56.4 | 52.5 | 72.4 | 60.5 | 29.4 | 54.6 | 54.3 |
| Ours-1.4B | 5.65 | 62.5 | 52.7 | 70.7 | 58.6 | 29.0 | 59.0 | 55.4 |
| GPT-Neo 2.7B | 5.63 | 62.2 | 55.8 | 72.1 | 61.1 | 30.2 | 57.6 | 56.5 |
| Hybrid H3-2.7B | 7.92 | 55.7 | 59.7 | 73.3 | 65.6 | 32.3 | 61.4 | 58.0 |
| OPT-2.7B | 5.12 | 63.6 | 60.6 | 74.8 | 60.8 | 31.3 | 61.0 | 58.7 |
| Pythia-2.8B | 5.04 | 64.7 | 59.3 | 74.0 | 64.1 | 32.9 | 59.7 | 59.1 |
| RWKV-3B | 5.24 | 63.9 | 59.6 | 73.7 | 67.8 | 33.1 | 59.6 | 59.6 |
| Ours-2.8B | 4.26 | 68.9 | 60.2 | 72.6 | 65.2 | 31.5 | 61.1 | 59.9 |
| GPT-J-6B | 4.10 | 68.3 | 66.3 | 75.4 | 67.0 | 36.6 | 64.1 | 63.0 |
| OPT-6.7B | 4.25 | 67.7 | 67.2 | 76.3 | 65.6 | 34.9 | 65.5 | 62.9 |
| Pythia-6.9B | 4.45 | 67.1 | 64.0 | 75.2 | 67.2 | 35.5 | 61.3 | 61.7 |
| RWKV-7.4B | 4.38 | 67.2 | 65.5 | 76.1 | 67.8 | 37.5 | 61.0 | 62.5 |

Table A4: Perplexity results on WikiText2 for LLaMA model family with 50% sparsity.

| Method | LLaMA-1-7B | LLaMA-1-13B | LLaMA-1-30B | LLaMA-2-7B | LLaMA-2-13B |
|---|---|---|---|---|---|
| / | 5.68 | 5.09 | 4.77 | 5.12 | 4.57 |
| Magnitude | 42.13 | 18.37 | 9.10 | 54.59 | 8.33 |
| SparseGPT 2:4 | 11.00 | 9.11 | 7.16 | 10.17 | 8.32 |
| Wanda 2:4 | 11.53 | 9.58 | 6.90 | 11.02 | 8.27 |
| Ours | **8.53** | **7.92** | **5.95** | **7.87** | **6.45** |

Table A5: Peak memory consumption results for Mamba models with different scales. The last column shows the percentage increase in memory usage for sparse models compared to dense models.

| Mamba | Dense | Sparse | **Increase (%)** |
|---|---|---|---|
| 130M | 708 MB | 725 MB | 2.4 |
| 370M | 1.64 GB | 1.68 GB | 2.4 |
| 790M | 3.28 GB | 3.39 GB | 3.3 |
| 1.4B | 5.22 GB | 5.42 GB | 3.8 |
| 2.8B | 7.61 GB | 8.03 GB | 5.5 |

Table A6: Energy consumption results for Mamba models with different model scales and sparsity ratios. ESR denotes energy saving ratio over the dense baseline.

| Mamba | Sparsity | Energy ( mW ) | ESR |
|---|---|---|---|
| | 0% | 1204.74 | - |
| 130M | 30% | 922.29 | 23.4% |
| | 50% | 689.60 | 42.8% |
| | 70% | 653.62 | 45.8% |
| | 0% | 1227.35 | - |
| 370M | 30% | 1109.43 | 9.6% |
| | 50% | 944.31 | 23.1% |
| | 70% | 925.03 | 24.7% |
| | 0% | 2030.38 | - |
| 790M | 30% | 1608.19 | 20.8% |
| | 50% | 1218.30 | 40.0% |
| | 70% | 1191.00 | 41.3% |
| | 0% | 2052.16 | - |
| 1.4B | 30% | 1897.14 | 7.6% |
| | 50% | 1843.56 | 10.2% |
| | 75% | 1783.54 | 13.1% |
| | 0% | 2763.21 | - |
| 2.8B | 30% | 2690.69 | 2.6% |
| | 50% | 2420.10 | 12.4% |
| | 75% | 2188.20 | 20.8% |

Table A7: Latency results of Mamba with different model scales and 64 sequence length, tested on a Xiaomi 6 device. SPD denotes the speedup over llama.cpp (red) and our dense baseline (blue).

| Mamba | Framework | Sparsity | Mobile CPU | | Mobile GPU | |
|---|---|---|---|---|---|---|
| | | | Token/s | SPD | Token/s | SPD |
| | llama.cpp | 0% | 0.73 | 1.0× | - | - |
| | ours | 0% | 2.90 | 4.0×/1.0× | 22.10 | 1.0× |
| 130M | ours | 30% | 3.00 | 4.1×/1.0× | 25.41 | 1.2× |
| | ours | 50% | 3.20 | 4.4×/1.1× | 28.52 | 1.3× |
| | ours | 70% | 3.47 | 4.8×/1.2× | 30.90 | 1.4× |
| | llama.cpp | 0% | 0.26 | 1.0× | - | - |
| | ours | 0% | 1.15 | 4.4×/1.0× | 11.91 | 1.0× |
| 370M | ours | 30% | 1.22 | 4.7×/1.1× | 12.50 | 1.1× |
| | ours | 50% | 1.48 | 5.7×/1.3× | 13.94 | 1.2× |
| | ours | 70% | 1.56 | 6.0×/1.4× | 15.79 | 1.3× |
| | llama.cpp | 0% | 0.12 | 1.0× | - | - |
| | ours | 0% | 0.40 | 3.3×/1.0× | 5.59 | 1.0× |
| 790M | ours | 30% | 0.49 | 4.1×/1.2× | 6.08 | 1.1× |
| | ours | 50% | 0.63 | 5.3×/1.6× | 6.76 | 1.2× |
| | ours | 70% | 0.72 | 6.0×/1.8× | 8.73 | 1.6× |