# OpenReview forum: "Sparse Learning for State Space Models on Mobile"
_ICLR.cc/2025/Conference — ICLR 2025 Poster_

### Official Review · Reviewer_iLUr · 2024-10-30

**Soundness:** 3
**Presentation:** 1
**Contribution:** 2
**Rating:** 6
**Confidence:** 4

**Summary:**

The paper introduces a solution for accelerating inference using the Mamba SSM on mobile devices. The solution includes a framework for pruning the model’s weights and an assortment of optimizations to improve inference execution, such as weight reordering, operator fusion, and layout transformation elimination. Experiments using different sizes of the Mamba model show that the presented sparsification method achieves better accuracy than other semi-structure pruning methods, namely SparseGPT and Wanda. The authors’ solution outperforms llama.cpp when running inference with Mamba on the CPU of a Snapdragon 8 Gen2 SoC, and exhibits further speedup on the GPU.

**Strengths:**

Overall, the sparse-learning framework for exploring pruning strategies for Mamba’s weights is promising. Furthermore, experimental results show an improvement in accuracy compared to other semi-structure pruning methods, SparseGPT and Wanda.

**Weaknesses:**

The paper presents an end-to-end solution, starting from a specific model, Mamba, pruning its weights, applying optimizations, and executing inference on a mobile SoC. Although such work must involve considerable engineering to implement the different stacks comprising the solution and gluing them together, it does not automatically produce any interesting results for the research-oriented community. Were there any particular challenges that had to be overcome, driving the development of specific innovations in the process? I do not see any such discussion in this submission. Although there can be innovations in the individual components, the full-stack presentation fails to sufficiently highlight them as, inevitably, we only get a rather shallow look into each one, both qualitatively and quantitatively. I am elaborating on that for every main component below.

The sparse-learning framework is very interesting, and the authors present improvements compared to other state-of-the-art pruning methods. However, the presentation is lacking. Are there other similar approaches to the Cn4 kernels? What is the related work here? SparseGPT and Wanda were initially developed for transformer-based models. Can the presented kernels tackle such models? If yes, how do they compare against the state of the art? If we are only looking at SSMs, why only Mamba? What about, e.g., S4 and H3?

The optimization workflow presented supposedly targets mobile devices. However, there is no explanation for why these optimizations are particularly good just for mobile devices and not all computational devices. The cited lack of “high throughput memory” (HBM?) on mobile devices is weak. All modern CPU and GPU architectures, both mobile and desktop/server, suffer from expensive data movement. Therefore, optimizations that reduce it are of general benefit. They may not offer equal benefit to all devices, but we cannot tell because there are no such comparisons in the paper. Maybe the optimizations take advantage of specific aspects of mobile architectures? There is no clear indication. There is a short discussion on SIMD units, but these are not fundamentally different between mobile and desktop/server devices.

There is a high-level description of the optimization workflow, but it is difficult to tell if any interesting innovations exist. Much prior work exists on optimizing sparse operations by reordering the non-zeros and introducing custom hierarchical sparse formats, such as the ParTI! Library (https://github.com/hpcgarage/ParTI). Other examples of prior work are DNNFusion (https://dl.acm.org/doi/10.1145/3453483.3454083) for operator fusion and SmartMem (https://dl.acm.org/doi/10.1145/3620666.3651384), which specifically addresses layout transformation elimination for mobile DNN execution. I am not saying that any of the above works are necessarily super relevant to the authors’ submission or that they need to be addressed, but if you are going to claim “a set of comprehensive compiler optimizations, including Cn4-specific optimizations and layout transformation elimination strategy on mobile devices” as a significant contribution, it will help to put your work into a better context.

The paper compares performance against llama.cpp. The authors provide an insight into why their solution is faster. Paraphrasing from their supplemental material, llama.cpp relies on a fixed pattern matching strategy to identify and fuse operation combinations, an approach that fails to recognize new combinations. Although llama.cpp is popular for executing transformer-based models, is it decent with SSMs? Aren’t there any better ways to execute Mamba and compare against them? If we are currently limited to llama.cpp because it is the only inference engine out there that currently supports Mamba and mobile, I would question if it is a “bad” baseline and how interesting the results are in the first place.

**Questions:**

- Is your sparse kernel design and sparse learning approach unique, or is there prior related work?
- Can your approach work for transformer-based models? If yes, have you done any accuracy comparisons?
- Have you tested your approach with other SSMs? If yes, do you have any accuracy comparisons?
- Why are your optimizations particularly good for mobile devices? Do you have any comparisons of the effects of your optimizations on mobile and desktop/server devices?
- Can you comment on the novelty of your optimizations compared to related work?
- Can you run your solution on non-mobile devices? If yes, do you have any results?
- How can your approach be extended to NPUs?
- You state in the paper that Mamba-370M achieves in your tests an average accuracy of 50.0%, while your 30%-sparse version achieves 50.6%. However, if one looks at the individual tests, there is no test where your version achieves higher accuracy than the original. Is there a typo?

---

> ### Author Response · Authors · 2024-11-25
> **Response to question 1-4**
>
> Response to Question 1:
>
> Compared with NVIDIA 2:4 pruning (and n:m pruning scheme in paper [1]),  the Cn4 kernel is motivated by the observation that modern mobile processors often feature Single Instruction Multiple Data (SIMD) units, which are optimized for parallel processing of data in groups, typically four elements at a time. By aligning the pruning pattern with the SIMD architecture, our method aims to maximize hardware utilization without specialized hardware support.  NVIDIA's 2:4 pruning, and more generally n:m pruning, typically fall under the category of structured pruning techniques. These techniques prune weights in a structured manner, removing entire groups of weights (e.g., rows, columns, filters) instead of individual weights. The paper's Cn4 kernel allows for adaptive sparsity, where the number of pruned weights (n) can vary across different kernels. This adaptability provides greater flexibility in tailoring the sparsity to the specific characteristics of the model and the data. Our methodology also emphasizes learning the optimal sparsity pattern through a dedicated sparse learning framework. This framework considers factors like accuracy loss, sparsity level, and latency constraints to guide the pruning process.
>
> Response to Question 2:
>
> We provide the perplexity results of LLaMA family on WikiText2 with 50% sparsity compared to the SparseGPT and Wanda in Table A4 at supplementary and also the following table. The results show that our method achieves better performance than the other two methods.
>
> | Method       | LLaMA-1-7B | LLaMA-1-13B | LLaMA-1-30B | LLaMA-2-7B | LLaMA-2-13B |
> |--------------|------------|-------------|-------------|------------|--------------|
> | /            | 5.68       | 5.09        | 4.77        | 5.12       | 4.57         |
> | Magnitude    | 42.13      | 18.37       | 9.10        | 54.59      | 8.33         |
> | SparseGPT 2:4| 11.00      | 9.11        | 7.16        | 10.17      | 8.32         |
> | Wanda 2:4    | 11.53      | 9.58        | 6.90        | 11.02      | 8.27         |
> | Ours         | **8.53**   | **7.92**    | **5.95**    | **7.87**   | **6.45**     |
>
>
> Response to Question 3:
>
> We have focused our exploration on SSMs within the NLP research domain, as their architectures are quite similar. Regarding SSMs in the computer vision domain, such as Vision-Mamba, we recognize the significance of this work and plan to apply our method to Vision-Mamba in future research endeavors.
>
> Response to Question 4:
>
> Our optimizations do specifically consider the unique architecture of mobile devices. For example, in the high-level architecture of Adreno GPUs (see page 15 in [2]), mobile devices feature two different memory units: buffer and image.  Buffers can only be read and written through the L2 cache, whereas images can be read through the L1 cache to achieve higher bandwidth but must still be written through the L2 cache.  Therefore, in our layout transformation elimination design, we account for this specific architecture, though we omitted detailed explanations due to paper length limitations. Since we primarily focus on mobile devices and design optimizations tailored to their architecture, we did not compare our results with server devices.
>
> [1] Learning N:M Fine-grained Structured Sparse Neural Networks From Scratch
>
> [2] Qualcomm® Snapdragon™ Mobile Platform OpenCL General Programming and Optimization

---

> ### Author Response · Authors · 2024-11-25
> **Response to question 5-8**
>
> Response to Question 5:
>
> Thanks for pointing this out. Our work focuses on accelerating State Space Models (SSMs), specifically the Mamba model, on resource-constrained mobile devices. We achieve this by introducing sparsity into the model's weight matrices, reducing computational complexity and memory usage. At the algorithm level, we propose a novel fine-grained structure pruning scheme called Cn4 Kernel Design, which offers high accuracy due to its flexibility and precision. We utilize a Sparse Learning Framework and Weight Compensation to further enhance the model's sparsity and accuracy. At the system/compiler level, we design Weight Reordering to benefit from regular memory access patterns and improve cache hit rates (increasing L1/L2/L3 cache hits by 5%, 4%, and 7% respectively; details will be added in our evaluation section). Our Efficient Sparse Weight Storage reduces storage costs compared to traditional CSR format by 1.5x.
>
> In contrast, DNNFusion aims to accelerate deep neural network execution broadly without focusing on specific models or platforms. It applies to various DNNs including CNNs, RNNs, and Transformers. SmartMem introduces a framework that eliminates most layout transformations by enabling multiple operators to use the same tensor layout through strategic choices in layouts and operator implementations. However, neither approach focuses on sparse computation and cannot be directly applied to sparse Mamba models. We will clarify this in our revision.
>
> Response to Question 6:
>
> Our solution is specifically designed for mobile architectures, so we have not tested it on non-mobile devices. However, with appropriate adaptations, it could potentially be applied to non-mobile platforms, which we may explore in future work.
>
> Response to Question 7:
>
> Our approach can also be extended to NPUs, as our Cn4-specific storage format is compatible with various devices. However, since NPUs have different memory organizations, what needs to be done is to adopt specific layouts for different operators when eliminating layout transformations in NPUs.
>
> Response to Question 8:
>
> It is not a typo. By adopting calibration in our post-training pruning method, we effectively eliminate non-essential connections, streamlining the network to focus on the most relevant features. This refinement allows the pruned model to achieve slightly better results than the dense model.

---

> ### Author Response · Authors · 2024-11-25
> **Response to weakness**
>
> Thanks for the suggestion. Llama.cpp is currently the only publicly available inference engine supporting Mamba on mobile platforms, making it the most practical baseline for our comparisons. We are actively monitoring developments in mobile inference engines, including MNN-LLM, MLC-LLM, TVM, ollama, ExecuTorch, NCNN, PowerInfer, exLLaMa, fastLLM, and TinyChatEngine, yet none of them support Mamba on mobile platforms now. We agree that a broader comparison with additional baselines would strengthen the paper further. Meanwhile, we aim to show how our optimizations outperform llama.cpp while providing insights into fundamental differences in execution strategies. We hope this addresses your concerns and demonstrates the significance of our findings despite current limitations in baseline availability.

---

> ### Author Response · Authors · 2024-11-25
>
> Dear Reviewer,
>
> Thank you very much for your valuable questions and feedback. As the discussion period is coming to a close, we sincerely hope you’ve had the opportunity to review our detailed responses to your previous questions and comments. If you have any further inquiries or require additional clarification, please don’t hesitate to reach out. We will do our best to address them promptly before the discussion deadline.
>
> Thank you again for your time and consideration.
>
> Best regards,
> The Authors

---

> > ### Comment · Reviewer_iLUr · 2024-11-26
> > **Follow-up**
> >
> > I want to thank the reviewers for their responses and clarifications. I have a few follow-up comments and/or questions.
> >
> > Question 2: Thank you for the additional data. Did you also happen to measure the accuracy for the same tests (LLaMA family on WikiText2 with 50% sparsity)?
> >
> > Question 4: It would still be interesting to elaborate further on it in the supplementary material, which I believe has no page limit.
> >
> > Question 5: I am very interested in reading more about this in your final revision.
> >
> > Question 8: I understand why a sparsified model can provide better accuracy than a dense one. My original question relates to the fact that your numbers do not add up. For convenience, I am posting them below. Please note that the 30% sparse model does not have better accuracy than the dense one in any of the datasets. What kind of average do you use that gives it overall higher accuracy?
> >
> > |Model|LAMBDA| HellaSwag| PIQA| Arc-E| Arc-C| WinoGrade| Avg.|
> > |:--|--:|--:|--:|--:|--:|--:|--:|
> > |Dense|55.6| 46.5| 69.5| 55.1| 28.0| 55.3| 50.0|
> > |30% Sparse|54.6| 44.9| 68.8| 52.7| 27.5| 55.3| 50.6|
> >
> > A general comment: If this paper gets accepted, the authors should improve their related work section (Section 2). They should at least add a discussion regarding the main contribution: the Cn4 kernel design and sparse learning framework.

---

> > > ### Author Response · Authors · 2024-11-26
> > > **Author response to follow up**
> > >
> > > Thanks to the reviewer for the follow up comments and questions for us.
> > >
> > > For Question 2: Yes. We apply 50% sparsity for magnitude pruning and utilize the 2:4 sparse pattern (removing 2 out of every 4 consecutive weights) for both SparseGPT and Wanda, both of which achieve 50% sparsity.
> > >
> > > For Question 4: Thanks for the acknowledgement, we will elaborate further in our supplementary with detailed explanations in our revision.
> > >
> > > For Question 5: Thanks for your interest, we will write more about this in our revision.
> > >
> > > For Question 8: Apologies for the typo in the reported average accuracy of the dense model. We sincerely regret this oversight and thank the reviewer for the patience and bringing it to our attention. The correct results are as follows,
> > >
> > > | Model          | LAMBDA | HellaSwag | PIQA  | Arc-E | Arc-C | WinoGrade | Average       |
> > > |----------------|--------|-----------|-------|-------|-------|-----------|---------------|
> > > | Dense          | 55.6   | 46.5      | 69.5  | 55.1  | 28.0    | 55.3      | 51.6         |
> > > | 30% Sparsity   | 54.6   | 44.9      | 68.8  | 52.7  | 27.5  | 55.3      | 50.6         |
> > >
> > >
> > > For general comment: Thank you for the suggestion. We will include a more comprehensive discussion of related works, particularly those pertaining to the $C_n^4$​ kernel design and the sparse learning framework.

---

> ### Author Response · Authors · 2024-12-02
>
> Dear Reviewer,
>
> Thank you again for your valuable time and thoughtful feedback on our submission. As the deadline for the Author/Reviewer discussion is approaching, we would greatly appreciate it if you could let us know whether our responses have addressed your concerns. Your confirmation or any further feedback would help us refine our work more effectively.
>
> We are more than happy to provide additional clarifications or address any remaining questions you might have.
>
> Thank you once again for your effort and consideration.
>
> Best regards,
> The Authors

---

> > ### Comment · Reviewer_iLUr · 2024-12-03
> >
> > I want to thank the authors again for their responses. I do not have any further questions. In general, my initial evaluation stands. I am increasing my score in light of the added results with the LLaMa model family, showing that the presented approach applies successfully beyond Mamba.
> >
> > I also want to suggest that the authors carefully review the numerical results before publishing or uploading the paper to arXiv. It is understandable that, with such a large volume of data, errors can occur when copying results to the document. By the way, although the authors have updated Table 2, in lines 472-473, it is still written that "compared with the dense Mamba-370M model with an average accuracy of 50.0% ..."

---

> > > ### Author Response · Authors · 2024-12-04
> > >
> > > Thank you for taking the time to review our submission and for raising your score. We truly appreciate your thoughtful evaluation and the constructive feedback you provided.
> > >
> > > We’ve noted the typos you mentioned, and we are committed to addressing them thoroughly in our revision. Your insights are invaluable in helping us refine our work to ensure clarity and precision.

---

### Official Review · Reviewer_6SJA · 2024-10-30

**Soundness:** 3
**Presentation:** 3
**Contribution:** 3
**Rating:** 8
**Confidence:** 3

**Summary:**

The paper introduces a novel learning framework for state space models (SSMs) that emphasizes kernel sparsity to enhance performance on mobile devices. The approach optimizes for an ideal balance between sparsity, latency, and accuracy, enabling an efficient pruning strategy suited for mobile hardware. With a robust theoretical foundation, the framework accounts for sparsity levels and latency impact on accuracy. Further, the authors incorporate hardware efficiency through architecture-aware compiler optimizations, including weight reordering, sparse weight storage, and eliminating layout transformations, achieving state-of-the-art (SOTA) results in accuracy at comparable or higher sparsity levels.

**Strengths:**

* Focus on Mobile Efficiency: The paper addresses a critical need for efficient execution of SSMs on mobile devices by optimizing for kernel sparsity without sacrificing accuracy, a valuable contribution to resource-constrained deep learning.
* Strong Theoretical Foundation: The proposed framework’s theoretical backing adds credibility, ensuring that the optimizations in sparsity and latency are rigorously derived, rather than heuristically implemented.
* Effective Hardware-Aware Optimization: Integrating architecture-aware compiler optimizations to handle sparse weight storage and reordering is a practical enhancement, boosting hardware efficiency while maintaining performance. This integration yields improved results over SOTA methods, showcasing either better accuracy for the same sparsity or higher sparsity for the same accuracy.* * Performance Presentation: The authors provide strong quantitative support,

**Weaknesses:**

* More discussion of latency trade-offs relative to different levels of accuracy and sparsity on the same model would be valuable for considering real-world deployment.
* A minor improvement could be made by bolding or highlighting the best results in each column of Table 2 for easy reference.

**Questions:**

* Could additional metrics be included for deployment feasibility? Metrics like power consumption and memory consumption could provide more insights for real-world deployment on mobile devices.

---

> ### Author Response · Authors · 2024-11-25
> **Response to weaknesses and questions**
>
> Response to Weakness 1:
>
> We have included the latency results for 30% sparsity in Table 3 and also the following table. The accuracy performance under 30% sparsity is already shown in Table 1. With this level of sparsity, our framework achieves a 1.1x to 1.2x speedup on mobile devices compared to the dense model and a 3.5x to 4.7x speedup compared to llama.cpp, highlighting its efficiency in mobile deployment.
>
> | Mamba  | Framework  | Sparsity | Mobile CPU Token/s | Mobile CPU SPD | Mobile GPU Token/s | Mobile GPU SPD |
> |--------|------------|----------|--------------------|----------------|--------------------|----------------|
> | 130M   | llama.cpp  | 0%       | 3.5                | 1.0x           | -                  | -              |
> |        | ours       | 0%       | 11.2               | 3.2x/1.0x      | 33.6               | 1.0x           |
> |        | ours       | 30%      | 12.1               | 3.5x/1.1x      | 40.7               | 1.2x           |
> | 370M   | llama.cpp  | 0%       | 1.5                | 1.0x           | -                  | -              |
> |        | ours       | 0%       | 6.1                | 4.1x/1.0x      | 19.6               | 1.0x           |
> |        | ours       | 30%      | 6.9                | 4.6x/1.1x      | 21.5               | 1.1x           |
> | 790M   | llama.cpp  | 0%       | 0.7                | 1.0x           | -                  | -              |
> |        | ours       | 0%       | 3.1                | 4.5x/1.0x      | 12.8               | 1.0x           |
> |        | ours       | 30%      | 3.3                | 4.7x/1.1x      | 14.1               | 1.1x           |
> | 1.4B   | llama.cpp  | 0%       | 0.6                | 1.0x           | -                  | -              |
> |        | ours       | 0%       | 2.1                | 3.5x/1.0x      | 10.1               | 1.0x           |
> |        | ours       | 30%      | 2.5                | 4.1x/1.2x      | 10.9               | 1.1x           |
> | 2.8B   | llama.cpp  | 0%       | 0.4                | 1.0x           | -                  | -              |
> |        | ours       | 0%       | 1.9                | 4.0x/1.0x      | 7.7                | 1.0x           |
> |        | ours       | 30%      | 2.2                | 4.7x/1.2x      | 9.1                | 1.2x           |
>
>
> We have added additional hardware metrics, including memory usage and energy consumption, in Table A5, Table A6 in Appendix, and also the following tables, to provide a more comprehensive evaluation.
>
> Response to Weakness 2:
>
> Thanks for the suggestion, we revised this in our rebuttal revision.
>
> Response to Question 1:
>
> We add more hardware metrics, including memory and energy consumption in Table A5 and Table A6, respectively, and also the following tables. Our sparse models result in slightly decreasing in memory, varying between 2.4% and 5.5%, while achieving substantial energy savings of up to 45.8% and significant inference acceleration. This highlights the efficiency and practicality of our approach, particularly for energy-sensitive and performance-critical applications.
>
> Peak memory consumption results.
>
> | Mamba  | Sparse   | Dense  | Decrease (%) |
> |--------|---------|---------|--------------|
> | 130M   | 708MB   | 725MB   | 2.4          |
> | 370M   | 1.64GB  | 1.68GB  | 2.4          |
> | 790M   | 3.28GB  | 3.39GB  | 3.3          |
> | 1.4B   | 5.22GB  | 5.42GB  | 3.8          |
> | 2.8B   | 7.61GB  | 8.03GB  | 5.5          |
>
>
>
> Energy consumption results.
>
> | Mamba  | Sparsity | Energy (mW) | Energy Saving Ratio |
> |--------|----------|-------------|---------------------|
> | 130M   | 0%       | 1204.74     | -                   |
> |        | 30%      | 922.29      | 23.4%              |
> |        | 50%      | 689.60      | 42.8%              |
> |        | 70%      | 653.62      | 45.8%              |
> | 370M   | 0%       | 1227.35     | -                   |
> |        | 30%      | 1109.43     | 9.6%               |
> |        | 50%      | 944.31      | 23.1%              |
> |        | 70%      | 925.03      | 24.7%              |
> | 790M   | 0%       | 2030.38     | -                   |
> |        | 30%      | 1608.19     | 20.8%              |
> |        | 50%      | 1218.30     | 40.0%              |
> |        | 70%      | 1191.00     | 41.3%              |
> | 1.4B   | 0%       | 2052.16     | -                   |
> |        | 30%      | 1897.14     | 7.6%               |
> |        | 50%      | 1843.56     | 10.2%              |
> |        | 70%      | 1783.54     | 13.1%              |
> | 2.8B   | 0%       | 2763.21     | -                   |
> |        | 30%      | 2690.69     | 2.6%               |
> |        | 50%      | 2420.10     | 12.4%              |
> |        | 70%      | 2188.20     | 20.8%              |

---

> > ### Comment · Reviewer_6SJA · 2024-11-25
> >
> > Thanks for the additional results which have adequately addressed my comments. I have updated my score.

---

> > > ### Author Response · Authors · 2024-11-26
> > >
> > > Thank you for raising your score, we greatly appreciate your thoughtful feedback and recognition of our work.

---

### Official Review · Reviewer_gxfe · 2024-11-03

**Soundness:** 3
**Presentation:** 1
**Contribution:** 3
**Rating:** 6
**Confidence:** 3

**Summary:**

Recently, State Space Models (SSMs) are gaining attention in sequential modeling problems. However it comes with increasing computational complexity and bandwidth demands. This paper proposes a sparse learning framework with architecture-aware compiler optimizations. This proposal includes 1. optimized kernels, 2. sparsity or latency oriented learning framework that uses 1.

The paper is too abstract to provide sufficient insight to readers. It seems that the framework and the algorithm provided seems to be a reasonable contribution. hence the score of 5: marginally below the acceptance threshold.

**Strengths:**

As the on-device AI is an important topic w.r.t privacy issues, it is an important direction to explore.

The paper seems to build a framework that reduces the computational intensity of the SSMs (which already is lower computational complexity than currently dominant Attention) seems to be a good direction.

It seems that the paper is claiming to have combined a number of compiler optimizations making it an end-to-end solution over simply exploring an algorithm-only or compiler-only or HW-only solution.

**Weaknesses:**

The paper is very difficult to understand. It seems that there are a lot of abstract explanations without concreteness that makes it difficult. Maybe adding some figures of what is really happening might help. For example, the paper states "our kernel is designed as Cn4 , which removes n elements from every group with four adjacent weights." There may be multiple ways by which this could happen and the paper does not dive into details.

**Questions:**

* Is there any performance impact associated with Remark 5.2.

* How long does this "sparse learning take" it seems like the computation seems to be quite complicated and may prolong the "learning" by a big factor. Can you provide some data?

* It seems that the work includes some compiler optimization that itself could be large enough to account for a separate paper. Can you provide more details to the infrastructure used? Is this some open-source work? Is this published anywhere?

---

> ### Author Response · Authors · 2024-11-25
> **Response to weakness and questions**
>
> Response to Weakness:
>
> To make it more clear, we define our kernel design in Section 5.1, and discuss the rationality of this design. The weights are split into multiple groups and each group has four consecutive weights. Our kernel is designed as Cn4, which removes n elements (0<=n<=4) from each group, as we can achieve practical acceleration with our compiler optimization for each case of Cn4.
>
> Response to Question 1:
>
> In practice, we have to adopt the dampening technique in Remark 5.2 to ensure the computation of matrix inversion. Without the dampening technique, the error of unavailable matrix inversion happens quite often. We set the dampening ratio $\gamma$ to 0.001. We ablate the performance of different dampening ratios and the results are demonstrated in Figure A1 of Appendix. We test on Mamba-2.8B model with 50% sparsity and compute the perplexity on LAMBDA dataset. As observed, smaller dampening ratios typically result in better performance, and further reducing the dampening ratio only leads to marginal improvements. A dampening ratio $\gamma$ smaller than 0.001 still leads to difficulties of matrix inversion. So we set  the dampening ratio $\gamma$ to 0.001.
>
> Response to Question 2:
>
> Although the computations in Equation (6) and (7) may seem to be complex, we can avoid the matrix multiplication with $e_{q_i}$ and $M_i$ by selecting rows or columns from a matrix with reduced complexity, due to their specific formation.  For example, $e_{q_i}^T W$ just selects the corresponding rows of $W$. So most of matrix multiplications can be implemented very efficiently. The most computation intensive part is the matrix inversion, which does not cost too much with a small or moderate dimension in 3B models or smaller.
>
> In the context of sparse learning, the original model weights are kept frozen, while only the sparse masks for each linear layer are trained. We explore optimal pruning strategies for each layer, leveraging the proposed specialized kernel while adhering to constraints such as sparsity and latency. As highlighted in Section 6.1, the sparse learning process for Mamba-2.8B requires just 50 minutes on an A6000 GPU, demonstrating its efficiency and relatively low computational overhead.
>
> Response to Question 3:
>
> Our work focuses on accelerating State Space Models (SSMs) with software-hardware co-design, specifically the Mamba model, on resource-constrained mobile devices. We achieve this by introducing sparsity into the model's weight matrices, reducing computational complexity and memory usage. At the algorithm level, we propose a novel fine-grained structure pruning scheme called Cn4 Kernel Design, which offers high accuracy due to its flexibility and precision. We utilize a Sparse Learning Framework and Weight Compensation to further enhance the model's sparsity and accuracy. At the system/compiler level, we design Weight Reordering to benefit from regular memory access patterns and improve cache hit rates (increasing L1/L2/L3 cache hits by 5%, 4%, and 7% respectively; details will be added in our evaluation section). Our Efficient Sparse Weight Storage reduces storage costs compared to traditional CSR format by 1.5x.
>
> We developed our sparse acceleration optimizations using C++, Python, and OpenCL. Specifically, we use Python for sparse training and compiler weight storage optimizations, while C++ and OpenCL are used for code generation and GPU acceleration. This original work has not been submitted or accepted elsewhere. We appreciate the author's recognition of our contribution and will add a subsection in the evaluation to clarify this in the revision.

---

> ### Author Response · Authors · 2024-11-25
>
> Dear Reviewer,
>
> Thank you very much for your questions. Since the discussion will end soon, we sincerely hope that you have found time to check our detailed response to your previous questions/comments. If you have any further questions, please feel free to let us know. We will try our best to reply to you before the discussion deadline.
>
> Thanks
> Authors.

---

> ### Comment · Reviewer_gxfe · 2024-11-27
>
> I appreciate the detailed response from the authors. I am convinced that the work makes a significant stride compared to the current SOTA works. I have increased the score to 6. However, I recommend that authors make some amendments to the paper to make the paper easier to read + make the details (various optimizations and the design) less abstract.

---

> > ### Author Response · Authors · 2024-11-27
> >
> > Thank you for improving your score; we truly value your feedback and recognition of our efforts. We will provide more details about the optimizations and design in the revision, and revise the flow for easier understanding.

---

### Official Review · Reviewer_vUZD · 2024-11-03

**Soundness:** 3
**Presentation:** 3
**Contribution:** 4
**Rating:** 8
**Confidence:** 3

**Summary:**

This paper introduces a mobile-friendly solution for SSMs, targeted for on-device inference. Specifically, the authors propose a sparsification & pruning method of contiguous weights along with a reordering operator for efficient on-device execution. Last, they introduce a sample-efficient compensation algorithm that recovers any lost accuracy. Results showcase gains of from 3.2x to 7x without gradual accuracy degradation as a function of sparsity.

**Strengths:**

* The paper contributes both an algorithmic/architectural change and a system hardware component implementation for efficient on-device execution, across CPU and GPU backends, which is evaluated on device.
* There has been great effort put in the evaluation and comparing against various baseline methods and models.
* The proposed method, albeit involved, is straightforward and through the ablation we witness the importance of each component in the resulting accuracy.

**Weaknesses:**

* The proposed technique has only been applied to one SSM architecture (Mamba) and evaluated on a single high-tier device.
* The on-device ML literature is quite old and there have been various contributions from 2018 onwards, also focusing on LLMs (see [a,b]).
* The advertised gains are not quoted over the same accuracy threshold.

**Questions:**

### Evaluation

* In the introduction, the authors quote that "llama.cpp takes over 5.8s to generate a single token with Mamba-2.8B", However, in table 3 of the evaluation, a similar setup is quoted at 0.4 tokens/sec = 2.5s per token. Could the authors clarify the source of this discrepancy?
* Although the performance benefits seem impressive at first sight, the 7x gains quote is quite misleading, as it comes with significant accuracy degradation. For a sub-1% degradation, the speedup gain is not quoted on Table 3.
* Why does Table 3 miss GPU execution for llama.cpp?
* In Section 6.2, the average column seems to represent an non-weighted average across the datasets, thus not taking into consideration the size of each task. Is this correct?

### Omissions / Extensions

* Results have only been benchmarked on a single high-tier device with 16GB of memory. I am wondering how the proposed solutions work on lower-end devices that do not have these level of resources. At the same time, it would be very interesting to quantify the energy requirements of running such workloads on device.
* A comparison with a similarly-sized Transformer-based LM would greatly enhance the evaluation and put the gains into perspective. Table A3 partly accomplished this, but there is no quantification of on-device performance.
* Furthermore, it would be valuable to see how these gains compare with different compression methods, such as quantization for example.
* Would there be any limitations of the method being applied to other modalities, such as vision (see Vision Mamba [c])
* Since SSMs come with memory benefits, as quoted in §3, it would be important to highlight the peak memory consumption of the pruned models during inference.
* It would be very insightful to visualise the sparsity per block in the resulting model to see if there is some kind or pattern in the pruning dynamics.

### Questions

* How does the compiler select the best layout per operator for different target devices?
* What is the overhead during training of the models with the proposed method? Are there gains from pruning during training?

[a] Xu, J., Li, Z., Chen, W., Wang, Q., Gao, X., Cai, Q., & Ling, Z. (2024). On-device language models: A comprehensive review. arXiv preprint arXiv:2409.00088.
[b] Liu, Z., Zhao, C., Iandola, F., Lai, C., Tian, Y., Fedorov, I., ... & Chandra, V. (2024). Mobilellm: Optimizing sub-billion parameter language models for on-device use cases. arXiv preprint arXiv:2402.14905.
[c] Zhu, L., Liao, B., Zhang, Q., Wang, X., Liu, W., & Wang, X. (2024). Vision mamba: Efficient visual representation learning with bidirectional state space model. Forty-First International Conference on Machine Learning (ICML).

### Nitpicking

* Table 1: sparsity typo
* Numbering issue on remarks (i.e. missing remark 5.1)
* Section 6.4: LAMBDA -> LAMBADA

---

> ### Author Response · Authors · 2024-11-25
> **Response to weaknesses**
>
> Response to Weakness 1:
>
> The perplexity results of LLaMA family on WikiText2 with 50% sparsity compared to the SparseGPT and Wanda are added as Table A4 in supplementary and also in the following table,
> | Method|LLaMA-1-7B|LLaMA-1-13B|LLaMA-1-30B|LLaMA-2-7B|LLaMA-2-13B|
> |-|-|-|-|-|-|
> |/| 5.68| 5.09| 4.77| 5.12| 4.57|
> |Magnitude|42.13| 18.37|9.10|54.59|8.33|
> |SparseGPT 2:4|11.00|9.11|7.16|10.17| 8.32|
> |Wanda 2:4|11.53| 9.58|6.90|11.02|8.27|
> |Ours|**8.53**|**7.92**|**5.95**|**7.87**|**6.45**|
>
> We also conducted the evaluation on a low-end device, the Xiaomi 6, which is equipped with a Snapdragon 835 featuring an octa-core CPU and Adreno 540 GPU with 6GB of memory. The results are presented in Table A7 at supplementary, as well as the table below. Due to the memory limitations, we report the results for Mamba models with 130M, 370M, and 790M model scales. The results show similar trends: our dense version achieves a speedup ranging from 3.3x to 4.4x compared to llama.cpp on the mobile CPU. Meanwhile, compared to our dense version, our sparse method demonstrates considerable acceleration, achieving approximately 1.1x  to 1.2x speedup at 30% sparsity, 1.1x to 1.3x speedup at 50% sparsity, and 1.2x to 1.8x speedup at 70% sparsity. These results demonstrate that our proposed method is both compatible and efficient across different edge devices.
>
> | Mamba  | Framework  | Sparsity | Mobile CPU Token/s | Mobile CPU SPD    | Mobile GPU Token/s | Mobile GPU SPD |
> |--------|------------|----------|--------------------|-------------------|--------------------|----------------|
> | 130M   | llama.cpp  | 0%       | 0.73               | 1.0x              | -                  | -              |
> |        | ours       | 0%       | 2.90               | 4.0x/1.0x         | 22.10              | 1.0x           |
> |        | ours       | 30%      | 3.00               | 4.1x/1.0x         | 25.41              | 1.2x           |
> |        | ours       | 50%      | 3.20               | 4.4x/1.1x         | 28.52              | 1.3x           |
> |        | ours       | 70%      | 3.47               | 4.8x/1.2x         | 30.90              | 1.4x           |
> | 370M   | llama.cpp  | 0%       | 0.26               | 1.0x              | -                  | -              |
> |        | ours       | 0%       | 1.15               | 4.4x/1.0x         | 11.91              | 1.0x           |
> |        | ours       | 30%      | 1.22               | 4.7x/1.1x         | 12.50              | 1.1x           |
> |        | ours       | 50%      | 1.48               | 5.7x/1.3x         | 13.94              | 1.2x           |
> |        | ours       | 70%      | 1.56               | 6.0x/1.4x         | 15.79              | 1.3x           |
> | 790M   | llama.cpp  | 0%       | 0.12               | 1.0x              | -                  | -              |
> |        | ours       | 0%       | 0.40               | 3.3x/1.0x         | 5.59               | 1.0x           |
> |        | ours       | 30%      | 0.49               | 4.1x/1.2x         | 6.08               | 1.1x           |
> |        | ours       | 50%      | 0.63               | 5.3x/1.6x         | 6.76               | 1.2x           |
> |        | ours       | 70%      | 0.72               | 6.0x/1.8x         | 8.73               | 1.3x           |
>
>
> Response to Weakness 2:
>
> Thanks for pointing this out. We will discuss the on-device efforts as follows,
>
> “Recently, several frameworks for on-device LLMs have emerged, including Llama.cpp, Power-Infer, Power-Infer2, ExecuTorch, MediaPipe, OpenLLM, and BentoLLM. Llama.cpp offers an efficient plug-and-play library for large language model inference across various hardware platforms. Power-Infer (and Power-Infer2) are designed for consumer-level GPUs with offloading support when loading weights for LLMs. ExecuTorch, MediaPipe, OpenLLM, and BentoLLM facilitate the deployment of open-source LLMs as local APIs or OpenAI-compatible API endpoints optimized for high throughput and cloud efficiency. However, none of them support sparse Mamba on mobile devices.”
>
> Response to Weakness 3:
>
> We have included the latency results for 30% sparsity in Table 3. With this level of sparsity, our framework achieves a 1.1x to 1.2x speedup on mobile devices compared to the dense model and a 3.5x to 4.7x speedup compared to llama.cpp, highlighting its efficiency in mobile deployment.

---

> ### Author Response · Authors · 2024-11-25
> **Response to questions (evaluation)**
>
> Response to Question (Evaluation 1):
>
> There is a typo in the introduction; the value should be '2.5s' instead of ‘5.8s’. We will correct it in the revision.
>
> Response to Question (Evaluation 2):
>
> We have added the 30% sparsity results in Table 3 and also below to show the speedup for each 1% decrease in accuracy.
>
> | Mamba  | Framework  | Sparsity | Mobile CPU Token/s | Mobile CPU SPD | Mobile GPU Token/s | Mobile GPU SPD |
> |--------|------------|----------|--------------------|----------------|--------------------|----------------|
> | 130M   | llama.cpp  | 0%       | 3.5                | 1.0x           | -                  | -              |
> |        | ours       | 0%       | 11.2               | 3.2x/1.0x      | 33.6               | 1.0x           |
> |        | ours       | 30%      | 12.1               | 3.5x/1.1x      | 40.7               | 1.2x           |
> | 370M   | llama.cpp  | 0%       | 1.5                | 1.0x           | -                  | -              |
> |        | ours       | 0%       | 6.1                | 4.1x/1.0x      | 19.6               | 1.0x           |
> |        | ours       | 30%      | 6.9                | 4.6x/1.1x      | 21.5               | 1.1x           |
> | 790M   | llama.cpp  | 0%       | 0.7                | 1.0x           | -                  | -              |
> |        | ours       | 0%       | 3.1                | 4.5x/1.0x      | 12.8               | 1.0x           |
> |        | ours       | 30%      | 3.3                | 4.7x/1.1x      | 14.1               | 1.1x           |
> | 1.4B   | llama.cpp  | 0%       | 0.6                | 1.0x           | -                  | -              |
> |        | ours       | 0%       | 2.1                | 3.5x/1.0x      | 10.1               | 1.0x           |
> |        | ours       | 30%      | 2.5                | 4.1x/1.2x      | 10.9               | 1.1x           |
> | 2.8B   | llama.cpp  | 0%       | 0.4                | 1.0x           | -                  | -              |
> |        | ours       | 0%       | 1.9                | 4.0x/1.0x      | 7.7                | 1.0x           |
> |        | ours       | 30%      | 2.2                | 4.7x/1.2x      | 9.1                | 1.2x           |
>
>
> Response to Question (Evaluation 3):
>
> Llama.cpp does not support GPU inference for Mamba on mobile devices during evaluation, so we only include CPU results. Currently, llama.cpp still lacks support because some SSM kernels are not implemented on mobile platforms yet.
>
> Response to Question (Evaluation 4):
> Yes, we maintain the same accuracy presentation as other state-of-the-art works.

---

> ### Author Response · Authors · 2024-11-25
> **Response to questions (omissions/extensions)**
>
> Response to Question (Omissions/Extensions 1):
>
> Our method does not conflict with the quantization method, we can adopt GPTQ for direct weight quantization to reduce the model size. While for GPUs, the acceleration brought by activation quantization may not be that large compared to CPUs.
> We further benchmark the performance on other edge devices (Xiaomi 6 mobile phone, equipped with QualComm Adreno 540 GPU and 6GB of memory). The result of energy consumption is reported in Table A6 at supplementary and also the following table. Compared to the dense model, our sparse model demonstrates a significant energy saving ratio, varying from 2.6% to 45.8%. The reduction in energy consumption is primarily attributed to our hardware design, which optimizes the proposed framework to significantly reduce computational demands. An important observation is that energy consumption decreases as the sparsity increases, highlighting the efficiency of our method in leveraging sparsity to minimize resource usage.
>
> | Mamba  | Sparsity | Energy (mW) | Energy Saving Ratio |
> |--------|----------|-------------|---------------------|
> | 130M   | 0%       | 1204.74     | -                   |
> |        | 30%      | 922.29      | 23.4%              |
> |        | 50%      | 689.60      | 42.8%              |
> |        | 70%      | 653.62      | 45.8%              |
> | 370M   | 0%       | 1227.35     | -                   |
> |        | 30%      | 1109.43     | 9.6%               |
> |        | 50%      | 944.31      | 23.1%              |
> |        | 70%      | 925.03      | 24.7%              |
> | 790M   | 0%       | 2030.38     | -                   |
> |        | 30%      | 1608.19     | 20.8%              |
> |        | 50%      | 1218.30     | 40.0%              |
> |        | 70%      | 1191.00     | 41.3%              |
> | 1.4B   | 0%       | 2052.16     | -                   |
> |        | 30%      | 1897.14     | 7.6%               |
> |        | 50%      | 1843.56     | 10.2%              |
> |        | 70%      | 1783.54     | 13.1%              |
> | 2.8B   | 0%       | 2763.21     | -                   |
> |        | 30%      | 2690.69     | 2.6%               |
> |        | 50%      | 2420.10     | 12.4%              |
> |        | 70%      | 2188.20     | 20.8%              |
>
> Response to Question (Omissions/Extensions 2):
>
> We also present the results for the LLaMA model family in Table A4 of the supplementary materials and the table above, demonstrating that our algorithm is well-suited for the LLaMA family. However, our framework currently does not support hardware deployment for the LLaMA models, as this would require additional optimizations tailored to the transformer architecture. We recognize the significance of this limitation and plan to address it in future research efforts.
>
> | Method       | LLaMA-1-7B | LLaMA-1-13B | LLaMA-1-30B | LLaMA-2-7B | LLaMA-2-13B |
> |--------------|------------|-------------|-------------|------------|--------------|
> | /            | 5.68       | 5.09        | 4.77        | 5.12       | 4.57         |
> | Magnitude    | 42.13      | 18.37       | 9.10        | 54.59      | 8.33         |
> | SparseGPT 2:4| 11.00      | 9.11        | 7.16        | 10.17      | 8.32         |
> | Wanda 2:4    | 11.53      | 9.58        | 6.90        | 11.02      | 8.27         |
> | Ours         | **8.53**   | **7.92**    | **5.95**    | **7.87**   | **6.45**     |
>
>
> Response to Question (Omissions/Extensions 3):
>
> Quantization is fully compatible with our proposed method and can be leveraged for additional model compression. For example, post-training quantization techniques like GPTQ [1] can be applied directly to quantize the model weights. Subsequently, our method can be employed to produce a sparse model, enabling efficient acceleration on mobile devices with a 4-bit representation.
>
> Response to Question (Omissions/Extensions 4):
>
> There is no such limitations, our method is based on the linear layers, it can be extended to Vision Mamba.
>
> Response to Question (Omissions/Extensions 5):
>
> We show the peak memory consumption during inference in Table A5 at supplementary and also the following table. Notably, The sparse model requires only a slight increase in memory compared to the dense model, ranging from 2.4% to 5.5%, while delivering significantly faster inference. This demonstrates the efficiency and practicality of our approach.
>
> | Mamba  | Dense   | Sparse  | Increase (%) |
> |--------|---------|---------|--------------|
> | 130M   | 708MB   | 725MB   | 2.4          |
> | 370M   | 1.64GB  | 1.68GB  | 2.4          |
> | 790M   | 3.28GB  | 3.39GB  | 3.3          |
> | 1.4B   | 5.22GB  | 5.42GB  | 3.8          |
> | 2.8B   | 7.61GB  | 8.03GB  | 5.5          |
>
> Response to Question (Omissions/Extensions 6):
>
> In our experiments, we adopt the uniform sparsity ratio for each SSM block.
>
>
> [1] GPTQ: Accurate Post-Training Quantization for Generative Pre-trained Transformers

---

> ### Author Response · Authors · 2024-11-25
> **Response to questions (questions)**
>
> Response to Question (Question 1):
>
> The details about layout transformation elimination, as well as the process of searching for the optimal data layout for each operator are elaborated in Appendix B.2. Regarding different target devices, our proposed method is compatible with different devices. We focus on mobile devices because they have limited bandwidth. By eliminating the layout transformation operations and selecting optimal data layout, we can significantly reduce the cost of memory access, which is a major overhead in mobile devices.
>
> Response to Question (Question 2):
>
> We utilize sparse learning to train the sparse masks and determine the optimal pruning strategy, keeping the model's original weights frozen throughout the process. For Mamba-2.8B, sparse learning is highly efficient, requiring approximately 50 minutes on a single A6000 GPU. Once the optimal sparse masks for each layer are obtained, pruning is performed based on these masks. Following pruning, we apply a compensation method to enhance the performance of the pruned model. This compensation process is a post-training, forward-only procedure that does not involve any backward computation, ensuring simplicity and efficiency.

---

> ### Comment · Reviewer_vUZD · 2024-11-25
> **Reply to authors' rebuttal**
>
> Thank you for your detailed rebuttal. I appreciate the additional experiments and insights. I have now raised my score.

---

> > ### Author Response · Authors · 2024-11-25
> >
> > Thank you for your thoughtful feedback and for taking the time to review our rebuttal. We sincerely appreciate your kind words and are grateful for your acknowledgment of our additional experiments and insights. Your support and the updated score mean a lot to us—thank you again!

---

### Meta-Review · Area_Chair_PLyM · 2024-12-23

**Metareview:**

The paper proposes a sparse-learning framework for accelerating the inference of SSMs, particularly the Mamba architecture, on mobile devices. The authors introduce the $C_n^4$ Kernel Design, a pruning strategy aligned with the SIMD architecture common in mobile processors, which maximizes hardware utilization. This kernel design is integrated into a pipeline of optimizations, including weight reordering, sparse weight storage, and layout transformation elimination, to improve performance on resource-constrained (mobile) devices. Here is the summary of the key findings:

- The framework shows a speedup of 3.2$\times$-7$\times$ for particular SSM models (Mamba) across mobile CPUs and GPUs with moderate memory overhead (still within the range of acceptable overhead) and energy consumption reductions of up to 45.8\%.
- The proposed sparse-learning framework demonstrates improved performance over SparseGPT and Wanda for LLaMA models, validating its applicability beyond SSMs.
- The approach balances sparsity, accuracy, and latency through a structured pruning and compensation mechanism, ensuring negligible accuracy degradation.

Here are the strengths of the paper: (a) the paper addresses a critical problem in deploying efficient ML models on mobile devices, making the paper relevant for mobile devices, (b) the proposed $C_n^4$ kernel exploits the SIMD architecture for efficient sparse computations on mobile devices. (c) extensive evaluation covering multiple model scales (from 130M to 2.8B params) and platforms and benchmarks against SOTA pruning methods (SparseGPT and Wanda) and transformer models (Llama). (d) energy reductions which is very important for mobile devices. (e) hardware-aware compiler optimizations further improves the value of the paper.

The reviewers also raised some concerns about the paper: (a) certain aspects of the method, such as weight reordering and layout transformation elimination, have been explored in prior work. (b) because of too much information, some reviewers found the paper challenging to follow (better visualization would help). (c) While Llama.cpp serves as a baseline, it is unclear if this is the optimal inference engine for SSMs, raising questions about the comprehensiveness of the comparison (I am not sure if there is an optimal inference engine dedicated for SSMs). (d) focusing heavily on Mamba models and lack evaluation across other SSM architecture. (e) some reviewers mentioned that the impact of optimization on non-mobile platforms is not explored (however I think this is ok because the paper clearly targets mobile devices).

After reading the reviewers' comments and the rebuttal, I recommend **Accept** for this paper. The rebuttal addressed most of the reviewers concerns and some of the them raised their scores, which has a weight in my recommendation for this paper.

**Additional Comments On Reviewer Discussion:**

## Reviewers Comments

- Reviewer (gxfe) noted that the paper was abstract and difficult to follow, with limited number of figures and visualization aid. The same reviewer also requested clearer explanations of concepts like the kernel design and sparse learning framework.
- Reviewer (iLUr) mentioned some inconsistencies in the results (which they attributed this to reporting such a large volume of data).
- Reviewer (iLUr) also questioned the adequacy of the baselines, particularly the reliance on Llama.cpp, which may not be the most optimized engine for SSMs. They also asked for evaluations across additional architectures like S4 and H3.
- Reviewer (iLUr) requested a deeper discussion of related work and a better theoretical context for the contributions, particularly for the proposed kernel.

---

## Authors' Responses

- After rebuttal, Reviewer (gxfe) acknowledged that this work makes a significant stride compared to current SOTA and increased their scores. However, the rebuttal seems to be convincing for them and they increased their scores to 6.
- Authors acknowledged their mistakes in reporting some of the results and agreed to fix all the issues before the final submission.
- Authors mentioned that Llama.cpp is currently the only publicly available inference engine supporting Mamba (which I agree with them). After rebuttal, the Reviewer (iLUr) raised their score to 6 (because of additional results and the detailed discussion).


Note that the Reviewer (iLUr) suggested that the authors carefully review the numerical results and also improve the related work section, particularly the reviewer requested a discussion regarding the main contribution of the paper, particularly the kernel and the sparse learning framework. I agree with the reviewer and hope that the authors can address these points in their final revision.

---

### Decision · Program_Chairs · 2025-01-22

Accept (Poster)